# Implicit Regularization of Discrete Gradient Dynamics in Linear Neural Networks

**Gauthier Gidel**
Mila & DIRO
Université de Montréal

**Francis Bach**
INRIA & École Normale Supérieure
PSL Research University, Paris

**Simon Lacoste-Julien**[*]
Mila & DIRO
Université de Montréal

## Abstract

When optimizing over-parameterized models, such as deep neural networks, a large set of parameters can achieve zero training error. In such cases, the choice of the optimization algorithm and its respective hyper-parameters introduces biases that will lead to convergence to specific minimizers of the objective. Consequently, this choice can be considered as an implicit regularization for the training of over-parametrized models. In this work, we push this idea further by studying the discrete gradient dynamics of the training of a two-layer linear network with the least-squares loss. Using a time rescaling, we show that, with a vanishing initialization and a small enough step size, this dynamics sequentially learns the solutions of a reduced-rank regression with a gradually increasing rank.

## 1  Introduction

When optimizing over-parameterized models, such as deep neural networks, a large set of parameters leads to a zero training error. However they lead to different values for the test error and thus have distinct generalization properties. More specifically, Neyshabur [2017, Part II] argues that the choice of the optimization algorithm (and its respective hyperparameters) provides an implicit regularization with respect to its geometry: it biases the training, finding a particular minimizer of the objective.

In this work, we use the same setting as Saxe et al. [2018]: a regression problem with least-squares loss on a multi-dimensional output. Our prediction is made either by a linear model or by a two-layer linear neural network [Saxe et al., 2018]. We extend their work which covered the *continuous* gradient dynamics, to weaker assumptions as well as analyze the behavior of the *discrete* gradient updates

We show that with a vanishing initialization and a small enough step-size, the gradient dynamics of a two-layer linear neural network sequentially learns components that can be ranked according to a hierarchical structure whereas the gradient dynamics induced by the same regression problem but with a linear prediction model instead learns these components simultaneously, missing this notion of hierarchy between components. The path followed by the two-layer formulation actually corresponds to successively solving the initial regression problem with a growing low rank constraint which is also know as reduced-rank regression [Izenman, 1975]. Note that this notion of path followed by *the dynamics of a whole network* is different from the notion of path introduced by Neyshabur et al. [2015a] which corresponds to a path followed *inside* a fixed network, i.e., one corresponds to training dynamics whereas the other corresponds to the propagation of information inside a network.

To sum-up, in our framework, the path followed by the gradient dynamics of a two-layer linear network provides an implicit regularization that may lead to potentially better generalization properties. Our contributions are the following:

---

[*]CIFAR fellow, Canada CIFAR AI chair
Correspondance to the first author: `<firstname>.<lastname>@umontreal.ca`

- Under some assumptions (see Assumption 1), we prove that both the *discrete* and continuous gradient dynamics sequentially learn the solutions of a gradually less regularized version of reduced-rank regression (Corollary 2 and 3). Among the close related work, such result on implicit regularization regarding discrete dynamics is novel. For the continuous case, we weaken the standard commutativity assumption using perturbation analysis.

- We experimentally verify the reasonableness of our assumption and observe improvements in terms of generalization (matrix reconstruction in our case) using the gradient dynamics of the two-layer linear network when compared against the linear model.

## 1.1 Related Work

The implicit regularization provided by the choice of the optimization algorithm has recently become an active area of research in machine learning, putting lot of interest on the behavior of gradient descent on deep over-parametrized models [Neyshabur et al., 2015b, 2017, Zhang et al., 2017].

Several works show that gradient descent on *unregularized* problems actually finds a minimum norm solution with respect to a particular norm that drastically depends on the problem of interest. Soudry et al. [2018] look at a logistic regression problem and show that the predictor does converge to the max-margin solution. A similar idea has been developed in the context of matrix factorization [Gunasekar et al., 2017]. Under the assumption that the observation matrices commute, they prove that gradient descent on this non-convex problem finds the minimum nuclear norm solution of the reconstruction problem, they also conjecture that this result would still hold without the commutativity assumption. This conjecture has been later partially solved by Li et al. [2018] under mild assumptions (namely the restricted isometry property). This work has some similarities with ours, since both focus on a least-squares regression problem over matrices with a form of matrix factorization that induces a non convex landscape. Their problem is more general than ours (see Uschmajew and Vandereycken [2018] for an even more general setting) but they are showing a result of a different kind from ours: they focus on the properties of the limit solution the continuous dynamics whereas we show some properties on the whole dynamics (continuous *and* discrete), proving that it actually visits points *during* the optimization that may provide good generalization. Interestingly, both results actually share common assumptions such as a commutativity assumption (which is less restrictive in our case since it is always true in some realistic settings such as linear autoencoders), vanishing initialization and a small enough step size.

Nar and Sastry [2018] also analyzed the gradient descent algorithm on a least-squares linear network model as a discrete time dynamical system, and derived certain necessary (but not sufficient) properties of the local optima that the algorithm can converge to with a non-vanishing step size. In this work, instead of looking at the properties of the limit solutions, we focus on the path followed by the gradient dynamics and precisely caracterize the weights learned along this path.

Combes et al. [2018] studied the continuous dynamics of some non-linear networks under relatively strong assumptions such as the linear separability of the data. Conversely, in this work, we do not make such separability assumption on the data but focus on linear networks.

Finally, Gunasekar et al. [2018] compared the implicit regularization provided by gradient descent in deep linear *convolutional* and *fully connected* networks. They show that the solution found by gradient descent is the minimum norm for both networks but according to a different norm. In this work, the fact that gradient descent finds the minimum norm solution is almost straightforward using standard results on least-squares. But the path followed by the gradient dynamics reveals interesting properties for generalization. As developed earlier, instead of focusing on the properties of the solution found by gradient descent, our goal is to study the path followed by the *discrete* gradient dynamics in the case of a two-layer linear network.

Prior work [Saxe et al., 2013, 2014, Advani and Saxe, 2017, Saxe et al., 2018, Lampinen and Ganguli, 2019] studied the gradient dynamics of two-layer linear networks and proved a result similar to our Thm. 2. We consider Saxe et al. [2018] as the closest related work, we re-use their notion of *simple deep linear neural network*, that we call two-layer neural networks, and use some elements of their proofs to extend their results. However, note that their work comes from a different perspective: through a mathematical analysis of a simple non-linear dynamics, they intend to highlight *continuous* dynamics of learning where one observes the sequential emergence of hierarchically structured notions to explain the regularities in representation of human semantic knowledge. In this work, we

are also considering a two-layer neural network but with an optimization perspective. We are able to extend Saxe et al. [2018, Eq. 6 and 7] weakening the commutativity assumption considered in Saxe et al. [2018] using perturbation analysis. In §4.1, we test to what extent our weaker assumption holds. Our main contribution is to show a similar result on the *discrete* gradient dynamics, that is important in our perspective since we aim to study the dynamics of gradient descent. This result cannot be trivially extended from the result on the *continuous* dynamics. We provide details on the difficulties of the proof in §3.2.

## 2 A Simple Deep Linear Model

In this work, we are interested in analyzing a least-squares model with multi-dimensional outputs. Given a *finite* number $n$ of inputs $\boldsymbol{x}_i \in \mathbb{R}^d$, $1 \le i \le n$ we want to predict a *multi-dimensional outputs* $\boldsymbol{y}_i \in \mathbb{R}^p$, $1 \le i \le n$ with a *deep linear network* [Saxe et al., 2018, Gunasekar et al., 2018],

$$\text{Deep linear model:} \quad \hat{\boldsymbol{y}}^d(\boldsymbol{x}) := \boldsymbol{W}_L^\top \cdots \boldsymbol{W}_1^\top \boldsymbol{x}\,, \tag{1}$$

where $\boldsymbol{W}_1, \ldots, \boldsymbol{W}_L$ are learned through a MSE formulation with the least-squares loss $f$,

$$(\boldsymbol{W}_1^*, \ldots, \boldsymbol{W}_L^*) \in \underset{\substack{\boldsymbol{W}_l \in \mathbb{R}^{r_{l-1} \times r_l} \\ 1 \le l \le L}}{\arg\min} \frac{1}{2n} \|\boldsymbol{Y} - \boldsymbol{X} \boldsymbol{W}_1 \cdots \boldsymbol{W}_L\|_2^2 =: f(\boldsymbol{W}_1, \ldots, \boldsymbol{W}_L)\,, \tag{2}$$

where $r_0 = d$, $r_l \in \mathbb{N}$, $1 \le l \le L - 1$ and $r_L = p$, $\boldsymbol{X} \in \mathbb{R}^{n \times d}$ and $\boldsymbol{Y} \in \mathbb{R}^{n \times p}$ are such that,

$$\boldsymbol{X}^\top := (\boldsymbol{x}_1 \; \cdots \; \boldsymbol{x}_n) \quad \text{and} \quad \boldsymbol{Y}^\top := (\boldsymbol{y}_1 \; \cdots \; \boldsymbol{y}_n)\,, \tag{3}$$

are the *design matrices* of $(\boldsymbol{x}_i)_{1 \le i \le n}$ and $(\boldsymbol{y}_i)_{1 \le i \le n}$. The *deep linear model* (1) is a $L$-layer deep linear neural network where we see $\boldsymbol{h}_l := \boldsymbol{W}_l \cdots \boldsymbol{W}_1 \boldsymbol{x}$ for $1 \le l \le L - 1$ as the $l^{th}$ *hidden layer*. At first, since this deep linear network cannot represent more than a linear transformation, we could think that there is no reason to use a deeper representation $L = 1$. However, in terms of learning flow, we will see in §3 that for $L = 2$ this model has a completely different dynamics from $L = 1$.

Increasing $L$ may induce a low rank constraint when $r := \min\{r_l : 1 \le l \le L - 1\} < \min(d, p)$. In that case, (2) is equivalent to a reduced-rank regression,

$$\boldsymbol{W}^{k,*} \in \underset{\substack{\boldsymbol{W} \in \mathbb{R}^{p \times d} \\ \operatorname{rank}(\boldsymbol{W}) \le r}}{\arg\min} \frac{1}{2n} \sum_{i=1}^n \|\boldsymbol{Y} - \boldsymbol{X} \boldsymbol{W}\|_2^2\,. \tag{4}$$

These problems have explicit solutions depending on $\boldsymbol{X}$ and $\boldsymbol{Y}$ [Reinsel and Velu, 1998, Thm. 2.2].

Note that, in this work we are interested in the implicit regularization provided in the context of *over-parametrized models*, i.e., when $r > \min(p, d)$. In that case,

$$\{\boldsymbol{W}_1 \cdots \boldsymbol{W}_L \; : \; \boldsymbol{W}_l \in \mathbb{R}^{r \times l-1, r_l}, \, 1 \le l \le L\} = \mathbb{R}^{p \times d}\,.$$

## 3 Gradient Dynamics as a Regularizer

In this section we would like to study the *discrete* dynamics of the gradient flow of (2), i.e.,

$$\boldsymbol{W}_l^{(t+1)} = \boldsymbol{W}_l^{(t)} - \eta \nabla_{\boldsymbol{W}_l} f\big(\boldsymbol{W}_{[L]}^{(t)}\big) \qquad \boldsymbol{W}_l^{(0)} \in \mathbb{R}^{r_{l-1} \times r_l}, 1 \le l \le L\,, \tag{5}$$

where we use the notation $\boldsymbol{W}_{[L]}^{(t)} := (\boldsymbol{W}_1^{(t)}, \ldots, \boldsymbol{W}_L^{(t)})$. The quantity $\eta$ is usually called the *step-size*. In order to get intuitions on the discrete dynamics we also consider its respective *continuous* version,

$$\dot{\boldsymbol{W}}_l(t) = -\nabla_{\boldsymbol{W}_l} f\big(\boldsymbol{W}_{[L]}(t)\big) \qquad \boldsymbol{W}_l(0) \in \mathbb{R}^{r_{l-1} \times r_l}, \, 1 \le l \le L\,, \tag{6}$$

where for $1 \le l \le L$, $\dot{\boldsymbol{W}}_l(t)$ is the temporal derivative of $\boldsymbol{W}_l(t)$. Note that there is no step-size in the continuous time dynamics since it actually corresponds to the limit of (5) when $\eta \to 0$. The continuous dynamics may be more convenient to study because such differential equations may have closed form solutions. In §3.1, we will see that under reasonable assumptions it is the case for (6).

### 3.1 Continuous dynamics

**Linear model: $L = 1$.** We start with the study of the continuous linear model, its gradient is,

$$\nabla f(\boldsymbol{W}) = \boldsymbol{\Sigma}_x \boldsymbol{W} - \boldsymbol{\Sigma}_{xy}, \tag{7}$$

where $\boldsymbol{\Sigma}_{xy} := \frac{1}{n} \boldsymbol{X}^\top \boldsymbol{Y}$ and $\boldsymbol{\Sigma}_x := \frac{1}{n} \boldsymbol{X}^\top \boldsymbol{X}$. Thus, $\boldsymbol{W}(t)$ is the solution of the differential equation,

$$\dot{\boldsymbol{W}}(t) = \boldsymbol{\Sigma}_{xy} - \boldsymbol{\Sigma}_x \boldsymbol{W}(t), \quad \boldsymbol{W}(0) = \boldsymbol{W}_0. \tag{8}$$

**Proposition 1.** *For any $\boldsymbol{W}_0 \in \mathbb{R}^{d \times p}$, the solution to the linear differential equation* (8) *is*

$$\boldsymbol{W}(t) = e^{-t\boldsymbol{\Sigma}_x}(\boldsymbol{W}_0 - \boldsymbol{\Sigma}_x^\dagger \boldsymbol{\Sigma}_{xy}) + \boldsymbol{\Sigma}_x^\dagger \boldsymbol{\Sigma}_{xy}, \tag{9}$$

*where $\boldsymbol{\Sigma}_x^\dagger$ is the pseudoinverse of $\boldsymbol{\Sigma}_x$.*

This standard result on ODE is provided in §B.1. Note that when $\boldsymbol{W}_0 \to \boldsymbol{0}$ we have

$$\boldsymbol{W}(t) \underset{\boldsymbol{W}_0 \to 0}{\to} (\boldsymbol{I}_d - e^{-t\boldsymbol{\Sigma}_x})\boldsymbol{\Sigma}_x^\dagger \boldsymbol{\Sigma}_{xy}. \tag{10}$$

**Deep linear network: $L \geq 2$.** The study of the deep linear model is more challenging since for $L \geq 2$, the landscape of the objective function $f$ is non-convex. The gradient flow of (2) is

$$\nabla f_{\boldsymbol{W}_l}(\boldsymbol{W}_{[L]}) = \boldsymbol{W}_{1:l-1}^\top(\boldsymbol{\Sigma}_x \boldsymbol{W} - \boldsymbol{\Sigma}_{xy})\boldsymbol{W}_{l+1:L}^\top \quad \text{where} \quad \boldsymbol{W}_{i:j} := \boldsymbol{W}_i \cdots \boldsymbol{W}_j, \ 1 \leq l \leq L, \tag{11}$$

where we used the convention that $\boldsymbol{W}_{1,0} = \boldsymbol{I}_d$ and $\boldsymbol{W}_{L+1,L} = \boldsymbol{I}_p$. Thus (6) becomes

$$\dot{\boldsymbol{W}}_l(t) = \boldsymbol{W}_{1:l-1}(t)^\top(\boldsymbol{\Sigma}_{xy} - \boldsymbol{\Sigma}_x \boldsymbol{W}(t))\boldsymbol{W}_{l+1:L}(t)^\top, \quad \boldsymbol{W}_l(0) \in \mathbb{R}^{d \times p}, \quad 1 \leq l \leq L. \tag{12}$$

We obtain a *coupled* differential equation (12) that is harder to solve than the previous linear differential equation (8) due, at the same time, to its non-linear components and to the coupling between $\boldsymbol{W}_l$, $1 \leq l \leq L$. However, in the case $L = 2$, Saxe et al. [2018] managed to find an explicit solution to this coupled differential equation under the assumption that "perceptual correlation is minimal" ($\boldsymbol{\Sigma}_x = \boldsymbol{I}_d$).[2] In this work we extend Saxe et al. [2018, Eq. 7] (for $L = 2$) under weaker assumptions. More precisely, we do not require the covariance matrix $\boldsymbol{\Sigma}_x$ to be the identity matrix. Let $(\boldsymbol{U}, \boldsymbol{V}, \boldsymbol{D})$ be the SVD of $\boldsymbol{\Sigma}_{xy}$, our assumption is the following:

**Assumption 1.** *There exist two orthogonal matrices $\boldsymbol{U}$, $\boldsymbol{V}$ such that we have the joint decomposition,*

$$\boldsymbol{\Sigma}_x = \boldsymbol{U}(\boldsymbol{D}_x + \boldsymbol{B})\boldsymbol{U}^\top \quad and \quad \boldsymbol{\Sigma}_{xy} = \boldsymbol{U}\boldsymbol{D}_{xy}\boldsymbol{V}^\top, \tag{13}$$

*where $\boldsymbol{B}$ is such that $\|\boldsymbol{B}\|_2 \leq \epsilon$ and $\boldsymbol{D}_x$, $\boldsymbol{D}_{xy}$ are matrices only with diagonal coefficients. We note $\sigma_1 \geq \cdots \geq \sigma_{r_{xy}} > 0$ the singular values of $\boldsymbol{\Sigma}_{xy}$ and $\lambda_1, \ldots, \lambda_{r_x}$ the diagonal entries of $\boldsymbol{D}_x$.*

Since two matrices commute if and only if they are co-diagonalizable [Horn et al., 1985, Thm. 1.3.21], the quantity $\epsilon$ represent to what extend $\boldsymbol{\Sigma}_x$ and $\boldsymbol{\Sigma}_{xy}\boldsymbol{\Sigma}_{xy}^\top$ do not commute. Before solving (12) under Assump. 1, we describe some motivating examples where the quantity $\epsilon$ is small or zero:

- **Linear autoencoder**: If $\boldsymbol{Y}$ is set to $\boldsymbol{X}$ and $L = 2$, we recover a linear autoencoder: $\hat{\boldsymbol{x}}(\boldsymbol{x}) = \boldsymbol{W}_2^\top \boldsymbol{W}_1^\top \boldsymbol{x}$, where $\boldsymbol{h} := \boldsymbol{W}_1^\top \boldsymbol{x}$ is the *encoded* representation of $\boldsymbol{x}$,

$$\boldsymbol{\Sigma}_{xy}\boldsymbol{\Sigma}_{xy}^\top = \left(\frac{1}{n}\boldsymbol{X}^\top \boldsymbol{X}\right)^2 = \boldsymbol{\Sigma}_x^2. \qquad \text{Thus, } \boldsymbol{B} = 0. \tag{14}$$

  Note that this linear autoencoder can also be interpreted as a form of principal component analysis. Actually, if we initialize with $\boldsymbol{W}_1 = \boldsymbol{W}_2^\top$, the gradient dynamics exactly recovers the PCA of $\boldsymbol{X}$, which is closely related to the matrix factorization problem of Gunasekar et al. [2017]. See §A where this derivation is detailed.

- **Deep linear *multi*-class prediction:** In that case, $p$ is the number of classes and $\boldsymbol{y}_i$ is a one-hot encoding of the class with, in practice, $p \ll d$. The intuition on why we may expect $\|\boldsymbol{B}\|_2$ to be small is because $\mathrm{rank}(\boldsymbol{Y}) \ll \mathrm{rank}(\boldsymbol{X})$ and thus the matrices of interest only have to almost commute on a small space in comparison to the whole space, thus $\boldsymbol{B}$ would be close to 0. We verify this intuition by computing $\|\boldsymbol{B}\|_2$ for several classification datasets in Table 1.

- **Minimal influence of perceptual correlation:** $\boldsymbol{\Sigma}_x \approx \boldsymbol{I}_d$. It is the setting discussed by Saxe et al. [2018]. We compare this assumption for some classification datasets with our Assump. 1 in §4.1.

**An explicit solution for** $L = 2$. Under Assump. 1 and specifying the initialization, one can solve the matrix differential equation for $\epsilon = 0$ and then use perturbation analysis to assess how close the solution of (8) is to the closed form solution derived for $\epsilon = 0$. This result is summarized in the following theorem proved in §B.2.

**Theorem 1.** *When $L = 2$, under Assump. 1, if we initialize with $\boldsymbol{W}_1(0) = \boldsymbol{U} \operatorname{diag}(e^{-\delta_1}, \dots, e^{-\delta_p})\boldsymbol{Q}$ and $\boldsymbol{W}_2(0) = \boldsymbol{Q}^{-1} \operatorname{diag}(e^{-\delta_1}, \dots, e^{-\delta_d})\boldsymbol{V}^\top$ where $\boldsymbol{Q}$ is an arbitrary invertible matrix, then the solution of (12) can be decomposed as the sum of the solution for $\epsilon = 0$ and a perturbation term,*

$$\begin{cases} \boldsymbol{W}_1(t) = \boldsymbol{W}_1^0(t) + \boldsymbol{W}_1^\epsilon(t) & where \quad \boldsymbol{W}_1^0(t) := \boldsymbol{U} \operatorname{diag}\left(\sqrt{w_1(t)}, \dots, \sqrt{w_p(t)}\right)\boldsymbol{Q} \\ \boldsymbol{W}_2(t) = \boldsymbol{W}_1^0(t) + \boldsymbol{W}_2^\epsilon(t) & where \quad \boldsymbol{W}_2^0(t) := \boldsymbol{Q}^{-1} \operatorname{diag}\left(\sqrt{w_1(t)}, \dots, \sqrt{w_d(t)}\right)\boldsymbol{V}^\top \end{cases} \quad (15)$$

*where we have $c > 0$ such that $\|\boldsymbol{W}_i^\epsilon(t)\| \leq \epsilon \cdot e^{ct^2}$ and,*

$$w_i(t) = \frac{\sigma_i e^{2\sigma_i t - 2\delta_i}}{\lambda_i(e^{2\sigma_i t - 2\delta_i} - e^{-2\delta_i}) + \sigma_i}, \ 1 \leq i \leq r_{xy}, \ w_i(t) = \frac{e^{-2\delta_i}}{1 + 2e^{-\delta_i}\lambda_i t}, \ r_{xy} < i \leq r_x \quad (16)$$

*where $(\sigma_i)$ and $(\lambda_i)$ are defined is Assump. 1. Note that $\operatorname{rank}(\boldsymbol{\Sigma}_{xy}) := r_{xy} \leq \operatorname{rank}(\boldsymbol{\Sigma}_x) := r_x$.*

The main difficulty in this result is the perturbation analysis for which we use a consequence of Grönwall's inequality [Gronwall, 1919] (Lemma 4). The proof can be sketched in three parts: first showing the result for $\epsilon = 0$, then showing that in the case $\epsilon > 0$, the matrices $\boldsymbol{W}_1(t)/t$ and $\boldsymbol{W}_2(t)/t$ are bounded and finally use Lemma 4 to get the perturbation bound.

This result is more general than the one provided by Saxe et al. [2018] because it requires a weaker assumption than $\boldsymbol{\Sigma}_x = \boldsymbol{I}_d$ and $\epsilon = 0$. In doing so, we obtain a result that takes into account the influence of correlations of the input samples. Note that Thm. 1 is only valid if the initialization $\boldsymbol{W}_1(0)\boldsymbol{W}_2(0)$ has the same singular vectors as $\boldsymbol{\Sigma}_{xy}$. However, making such assumptions on the initialization is standard in the literature and, in practice, we can set the initialization of the optimization algorithm in order to also ensure that property. For instance, in the case of the linear autoencoder, one can set $\boldsymbol{W}_1(0) = \boldsymbol{W}_2(0) = e^{-\delta}\boldsymbol{I}_d$.

In the following subsection we will use Thm. 1 to show that the components $[\boldsymbol{U}]_i$, $1 \leq i \leq r_{xy}$ in the order defined by the decreasing singular values of $\boldsymbol{\Sigma}_{xy}$ are learned sequentially by the gradient dynamics.

**Sequential learning of components.** The sequential learning of the left singular vectors of $\boldsymbol{\Sigma}_{xy}$ (sorted by the magnitude of its singular values) by the *continuous* gradient dynamics of deep linear networks has been highlighted by Saxe et al. [2018]. They note in their Eq. (10) that the $i^{th}$ phase transition happens approximately after a time $T_i$ defined as (using our notation),

$$T_i := \frac{\delta_i}{\sigma_i} \ln(\sigma_i) \quad where \quad \boldsymbol{\Sigma}_{xy} = \sum_{i=1}^{r_{xy}} \sigma_i \boldsymbol{u}_i \boldsymbol{v}_i^\top . \quad (17)$$

They argue that as $\delta_i \to \infty$, the time $T_i$ is roughly $O(1/\sigma_i)$. The intuition is that a vanishing initialization increases the gap between the phase transition times $T_i$ and thus tends to separate the learning of each components. However, a vanishing initialization just formally leads to $T_i \to \infty$.

In this work, we introduce a notion of *time rescaling* in order to formalize this notion of phase transition and we show that, after this time rescaling, the point visited between two phase transitions is the solution of a low rank regularized version (4) of the initial problem (2) with the low rank constraint that loosens sequentially.

The intuition behind time rescaling is that it counterbalances the vanishing initialization in (17): Since $T_i$ grows as fast as $\delta_i$ we need to multiply the time by $\delta_i$, in order to grow at the same pace as $T_i$.

Using this rescaling we can present our theorem, proved in §B.3, which says that a vanishing initialization tends to force the sequential learning of the component of $\boldsymbol{X}$ associated with the largest singular value of $\boldsymbol{\Sigma}_{xy}$. Note that we need to rescale the time *uniformly* for each component. That is why in the following we set $\delta_i = \delta$, $1 \leq i \leq \max(p, d)$.

**Theorem 2.** *Let us denote $w_i(t)$, the values defined in (16). If $w_i(0) = e^{-\delta}$, $1 \leq i \leq r$, and $\epsilon = e^{-\delta^2 \ln(\delta)}$ then we have that $w_i(\delta t)$ converge to a step function as $\delta \to \infty$:*

$$w_i(\delta t) \underset{\delta \to \infty}{\to} \frac{\sigma_i}{\lambda_i + \sigma_i}\mathbb{1}\{t = T_i\} + \frac{\sigma_i}{\lambda_i}\mathbb{1}\{t > T_i\} . \quad (18)$$

*where $T_i := 1/\sigma_i$, $\mathbb{1}\{t \in A\} = 1$ if $t \in A$ and $0$ otherwise.*

Notice how the $i^{th}$ components of $\boldsymbol{W}_1$ and $\boldsymbol{W}_2$ are inactive, i.e., $w_i(t)$ is zero, for small $t$ and is suddenly learned when $t$ reaches the phase transition time $T_i := 1/\sigma_i$. As shown in Prop. 1 and illustrated in Fig. 1, this sequential learning behavior does not occur for the non-factorized formulation. Gunasekar et al. [2017] observed similar differences between their factorized and not factorized formulations of matrix regression. Note that, the time rescaling we introduced is $t \to \delta t$, in order to compensate the vanishing initialization, rescaling the time and taking the limit this way for (8) would lead to a constant function.

Gunasekar et al. [2017] also had to consider a vanishing initialization in order to show that on a simple matrix factorization problem the continuous dynamics of gradient descent does converge to the minimum nuclear norm solution. This assumption is necessary in such proofs in order to avoid to initialize with wrong components. However one cannot consider an initialization with the null matrix since it is a stationary point of the dynamics, that is why this notion of double limit (vanishing initialization and $t \to \infty$) is used.

From Thm. 2, two corollaries follow directly. The first one regards the nuclear norm of the product $\boldsymbol{W}_1(\delta t)\boldsymbol{W}_2(\delta t)$. This corollary says that $\|\boldsymbol{W}_1(\delta t)\boldsymbol{W}_2(\delta t)\|_*$ is a step function and that each increment of this integer value corresponds to the learning of a new component of $\boldsymbol{X}$. These components are leaned by order of relevance, i.e., by order of magnitude of their respective eigenvalues and the learning of a new component can be easily noticed by an incremental gap in the nuclear norm of the matrix product $\boldsymbol{W}_1(\delta t)\boldsymbol{W}_2(\delta t)$,

**Corollary 1.** *Let $\boldsymbol{W}_1(t)$ and $\boldsymbol{W}_2(t)$ be the matrices solution of (12) defined in (15). The limit of the squared euclidean norm of $\boldsymbol{W}_1(t)\boldsymbol{W}_2(t)$ when $\delta \to \infty$ is a step function defined as,*

$$\|\boldsymbol{W}_1(\delta t)\boldsymbol{W}_2(\delta t)\|_2^2 \underset{\delta \to \infty}{\to} \sum_{i=1}^{r_{xy}} \frac{\sigma_i^2}{\lambda_i^2}\mathbb{1}\{T_i < t\} + \frac{\sigma_i^2}{(\lambda_i+\sigma_i)^2}\mathbb{1}\{T_i = t\} \tag{19}$$

*where $T_i := 1/\sigma_i$ and $\sigma_1 > \cdots > \sigma_{r_{xy}} > 0$ are the positive singular values of $\boldsymbol{\Sigma}_{xy}$.*

It is natural to look at the norm of the product $\boldsymbol{W}_1(\delta t)\boldsymbol{W}_2(\delta t)$ since in Thm. 2, $(w_i(t))$ are its singular values. However, since the rank of $\boldsymbol{W}_1(\delta t)\boldsymbol{W}_2(\delta t)$ is discontinuously increasing after each phase transition, any norm would jump with respect to the rank increments. We illustrate these jumps in Fig. 1 where we plot the closed form of the squared $\ell_2$ norms of $t \mapsto \boldsymbol{W}(\delta t)$ and $t \mapsto \boldsymbol{W}_1(\delta t)\boldsymbol{W}_2(\delta t)$ for vanishing initializations with $\boldsymbol{\Sigma}_{yx} = \mathrm{diag}(10^{-1}, 10^{-2}, 10^{-3})$ and $\boldsymbol{\Sigma}_x = \boldsymbol{I}_d$.

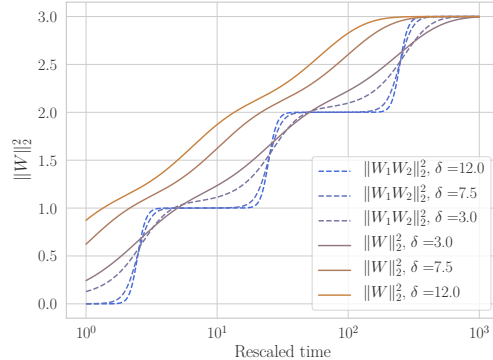

Figure 1: Closed form solution of squared $\ell_2$ norm of $\boldsymbol{W}(\delta t)$ and $\boldsymbol{W}_1(\delta t)\boldsymbol{W}_2(\delta t)$ respectively for a linear model and a two-layer linear autoencoder, depending on $\boldsymbol{W}(0) = \boldsymbol{W}_1(0)\boldsymbol{W}_2(0) = e^{-\delta}\boldsymbol{I}_d$. Note that for an autoencoder $\lambda_i = \sigma_i$ and thus the trace norm has integer values. According to Thm. 2, the integer trace norm increment represents the learning of a new component.

From Thm. 2, we can notice that, between time $T_k$ and $T_{k+1}$, the rank of the limit matrix $\boldsymbol{W}_1\boldsymbol{W}_2$ is actually equal to $k$, meaning that at each phase transition, the rank of $\boldsymbol{W}_1\boldsymbol{W}_2$ is increased by 1. Moreover, this matrix product contains the $k$ components of $\boldsymbol{X}$ corresponding to the $k$ largest singular values of $\boldsymbol{\Sigma}_{xy}$. Thus, we can show that this matrix product is the solution of the $k$-low rank constrained version (4) of the initial problem (2),

**Corollary 2.** *Let $\boldsymbol{W}_1(t)$ and $\boldsymbol{W}_2(t)$ be the matrices solution of (12) defined in (15). We have that,*

$$\frac{1}{\sigma_k} < t < \frac{1}{\sigma_{k+1}} \quad \Rightarrow \quad \boldsymbol{W}_1(\delta t)\boldsymbol{W}_2(\delta t) \underset{\delta \to \infty}{\to} \boldsymbol{W}^{k,*}, \qquad 1 \le k \le r_{xy}. \tag{20}$$

*where $\boldsymbol{W}^{k,*}$ is the minimum $\ell_2$ norm solution of the reduced-rank-$k$ regression problem (4) .*

## 3.2 Discrete dynamics

We are interested in the behavior of optimization methods. Thus, the gradient dynamics of interest is the *discrete* one (5). A major contribution of our work is thus contained in this section. The continuous case studied in § 3.1 provided good intuitions and insights on the behavior of the potential discrete dynamics that we can use for our analysis.

**Why the discrete analysis is challenging.** Previous related work [Gunasekar et al., 2017, Saxe et al., 2018] only provide results on the continuous dynamics. Their proofs use the fact that their respective continuous dynamics of interest have a closed form solution (e.g., Thm.1). To our knowledge, no closed form solution is known for the discrete dynamics (5). Thus its analysis requires a new proof technique. Moreover, using Euler's integration methods, one can show that both dynamics are close but only for a vanishing step size depending on a *finite* horizon. Such dependence on the horizon is problematic since the time rescaling used in Thm. 2 would make any finite horizon go to infinity. In this section, we consider realistic conditions on the step-size (namely, it has to be smaller than the Lipschitz constant and some notion of eigen-gap) without any dependence on the horizon. Such assumption is relevant since we want to study the dynamics of practical optimization algorithms (i.e., with a step size as large as possible and without knowing in advance the horizon).

**Single layer linear model.** In this paragraph, we consider the discrete update for the linear model. Since $L = 1$, for notational compactness, we call $\boldsymbol{W}_t$ the matrix updated according to (5). Using the gradient derivation (7), the discrete update scheme for the *linear model* is,

$$\boldsymbol{W}_{t+1} = \boldsymbol{W}_t - \eta(\boldsymbol{\Sigma}_x \boldsymbol{W}_t - \boldsymbol{\Sigma}_{xy}) = (\boldsymbol{I}_d - \eta\boldsymbol{\Sigma}_x)\boldsymbol{W}_t + \eta\boldsymbol{\Sigma}_{xy} \,.$$

Noticing that for $1/\lambda_{\max}(\boldsymbol{\Sigma}_x) > \eta > 0$, $\boldsymbol{I}_d - \eta\boldsymbol{\Sigma}_x$ is invertible, this recursion (see §B.4) leads to,

$$\boldsymbol{W}_t = (\boldsymbol{W}_0 - \boldsymbol{\Sigma}_x^\dagger \boldsymbol{\Sigma}_{xy})(\boldsymbol{I}_d - \eta\boldsymbol{\Sigma}_x)^t + \boldsymbol{\Sigma}_x^\dagger \boldsymbol{\Sigma}_{xy} \,. \tag{21}$$

We obtain a similar result as the solution of the differential equation given in Prop. 1. With a vanishing initialization we reach a function that does not sequentially learn some components.

**Two-layer linear model.** The discrete update scheme for the *two-layer linear network* (2) is,

$$\boldsymbol{W}_1^{(t+1)} = \boldsymbol{W}_1^{(t)} - \eta(\boldsymbol{\Sigma}_x \boldsymbol{W}^{(t)} - \boldsymbol{\Sigma}_{xy})(\boldsymbol{W}_2^{(t)})^\top \,, \ \ \boldsymbol{W}_2^{(t+1)} = \boldsymbol{W}_2^{(t)} - \eta(\boldsymbol{W}_1^{(t)})^\top(\boldsymbol{\Sigma}_x \boldsymbol{W}^{(t)} - \boldsymbol{\Sigma}_{xy}) \,.$$

When $\epsilon = 0$, by a change of basis and a proper initialization, we can reduce the study of this matrix equation to $r$ independant dynamics (see §B.5 for more details), for $1 \le i \le r$,

$$w_i^{(t+1)} = w_i^{(t)} + \eta w_i^{(t)}(\sigma_i - \lambda_i w_i^{(t)} w_i^{(t)}) \,. \tag{22}$$

Thus we can derive a bound on the iterate $w_i^{(t)}$ leading to the following theorem,

**Theorem 3.** *Under the same assumptions as Thm. 1 and $\epsilon = 0$, we have*

$$\boldsymbol{W}_1^{(t)} = \boldsymbol{U} \operatorname{diag}\left(\sqrt{w_1^{(t)}}, \dots, \sqrt{w_p^{(t)}}\right)\boldsymbol{Q} \quad and \quad \boldsymbol{W}_2^{(t)} = \boldsymbol{Q}^{-1} \operatorname{diag}\left(\sqrt{w_1^{(t)}}, \dots, \sqrt{w_d^{(t)}}\right)\boldsymbol{V}^\top \,.$$

*Moreover, for any $1 \le i \le r_{xy}$, if $1 > w_i^{(0)} > 0$ and $2\eta\sigma_i < 1$, then $\forall t \ge 0$, $1 \le i \le r_x$ we have,*

$$\frac{w_i^{(0)}}{(\sigma_i - \lambda_i w_i^{(0)})e^{(-2\eta\sigma_i + 4\eta^2\sigma_i^2)t} + w_i^{(0)}\lambda_i} \le w_i^{(t)} \le \frac{w_i^{(0)}}{(\sigma_i - \lambda_i w_i^{(0)})e^{(-2\eta\sigma_i - \eta^2\sigma_i^2)t} + w_i^{(0)}\lambda_i} \,, \tag{23}$$

*and $w_i^{(t)} \le \frac{w_i^{(0)}}{1+w_i^{(0)}\lambda_i \eta t}$ for $r_{xy} \le i \le r_x$. The differences with the continuous case (16) are in red.*

*Proof sketch.* The solution of the continuous dynamics lets us think directly studying the sequence $w_i^{(t)}$ might be quite challenging since the solution of the continuous dynamics $w_i(t)^{-1}$ has a non-linear and non-convex behavior.

The main insight from this proof is that one can treat the discrete case using the right transformation, to show that some sequence doee converge linearly.

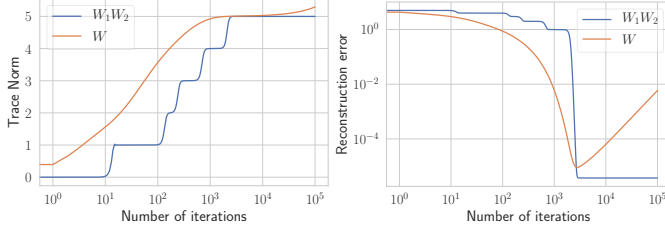

| Dataset | $\Delta_{xy}$ | $\Delta_x$ |
|---|---|---|
| MNIST | $2.8 \times 10^{-2}$ | .70 |
| CIFAR-10 | $3.0 \times 10^{-2}$ | .68 |
| ImageNet | $1.7 \times 10^{-1}$ | .70 |

Figure 2: Trace norm and reconstruction errors of $\boldsymbol{W}^{(t)}$ for $L = 1$ and 2 as a function of $t$.

Table 1: Value of the quantities $\Delta_{xy}$ and $\Delta_x$ defined in (27).

Thm. 2 indicates the quantity $w_i(t)^{-1} - \frac{\lambda_i}{\sigma_i}$ is the good candidate to show linear convergence to 0,

$$w_i(t)^{-1} - \frac{\sigma_i}{\lambda_i} = (w_i(0)^{-1} - \frac{\sigma_i}{\lambda_i})e^{-2\eta\sigma_i t} . \tag{24}$$

What we can expect is thus to show that the sequence $(w_i^{(t)})^{-1} - \frac{\sigma_i}{\lambda_i}$ has similar properties. The first step of the proof is to show that $(w_i^{(t)})$ is an increasing sequence smaller than one. The second step is then to use (22) to get,

$$\frac{1}{w_i^{(t+1)}} - \frac{\lambda_i}{\sigma_i} = \frac{1}{w_i^{(t)}}\left(\frac{1}{1+2(\sigma_i-\lambda_i w_i^{(t)})+\eta^2(\sigma_i-\lambda_i w_i^{(t)})^2}\right) - \frac{\lambda_i}{\sigma_i}$$

Then using that $1 - x \le \frac{1}{1+x} \le 1 - x + x^2$ for any $1 \le x \le 0$ we can derive the upper and lower bounds on the linear convergence rate. See §B.5 for full proof. □

In order to get a similar interpretation of Thm. 3 in terms of implicit regularization, we use the intuitions from Thm. 2. The analogy between continuous and discrete time is that the discrete time dynamics is doing $t$ time-steps of size $\eta$, meaning that we have $\boldsymbol{W}(\eta t) \approx \boldsymbol{W}_t$, the time rescaling in continuous time consists in multiplying the time by $\delta$ thus we get the analog phase transition time,

$$\eta T_i := \frac{1}{\sigma_i} \quad \Rightarrow \quad T_i := \frac{1}{\eta\sigma_i} . \tag{25}$$

Recall that we assumed that $m_i^{(0)} = n_i^{(0)} = e^{-\delta}$. Thus, we show that the $i^{th}$ component is learned around time $T_i$, and consequently that the components are learned sequentially,

**Corollary 3.** *If* $\eta < \frac{1}{2\sigma_1}$, $\eta < 2\frac{\sigma_i-\sigma_{i+1}}{\sigma_i^2}$ *and* $\eta < \frac{\sigma_i-\sigma_{i+1}}{2\sigma_{i+1}^2}$, *for* $1 \le i \le r_{xy}-1$, *then for* $1 \le i < r_x$,

$$w_i^{(\delta T_j)} \underset{\delta\to\infty}{\to} \begin{cases} 0 & \text{if } i > r_{xy} \quad \text{or} \quad j < i \\ \frac{\sigma_i}{\lambda_i} & \text{if } i \le r_{xy} \quad \text{and} \quad j > i . \end{cases} \tag{26}$$

*where* $T_0 := 0$, $T_j := \frac{1}{\sigma_j\eta}$, $1 \le j \le r_{xy}$ *and* $T_j := +\infty$ *if* $j > r_{xy}$.

This result is proved in §B.5. The quantities $\frac{\sigma_i-\sigma_{i+1}}{\sigma_i^2}$ and $\frac{\sigma_i-\sigma_{i+1}}{\sigma_{i+1}^2}$ can be interpreted as relative *eigen-gaps*. Note that they are well defined since we assumed that the eigenspaces were unidimensional. The intuition behind this condition is that the step-size cannot be larger than the eigen-gaps because otherwise the discrete optimization algorithm would not be able to distinguish some components.

## 4 Experiments

### 4.1 Assump. 1 for Classification Datasets

In this section we verify to what extent Assump. 1 is true on standard classification datasets. For this, we compute the normalized quantities $\Delta_{xy}$ and $\Delta_x$ representing how much Assump. 1 and the assumption that $\boldsymbol{\Sigma}_x \approx \boldsymbol{I}_d$ are respectively broken. We compute $\boldsymbol{B}$ by computing $\boldsymbol{U}$, the left singular vector of $\boldsymbol{\Sigma}_{xy}$ and looking at the non-diagonal coefficients of $\boldsymbol{U}^\top\boldsymbol{\Sigma}_x\boldsymbol{U}$,

$$\Delta_{xy} := \frac{\|\boldsymbol{B}\|_2}{\|\boldsymbol{\Sigma}_x\|_2}, \quad \Delta_x := \frac{1}{2}\left\|\hat{\boldsymbol{\Sigma}}_x - \hat{\boldsymbol{I}}_d\right\|_2 , \tag{27}$$

where $\|\cdot\|$ is the Frobenius norm, the $\hat{\mathbf{\Sigma}}$ expressions correspond to $\hat{\boldsymbol{X}} := \boldsymbol{X}/\|\boldsymbol{X}\|$ and $\hat{\boldsymbol{I}}_d := \boldsymbol{I}_d/\|\boldsymbol{I}_d\|$. These normalized quantities are between $0$ and $1$. The closer to $1$, the less the assumption hold and conversely, the closer to $0$, the more the assumption approximately holds. We present the results on three standard classification datasets, MNIST [LeCun et al., 2010], CIFAR10 [Krizhevsky et al., 2014] and ImageNet [Deng et al., 2009], a down-sampled version of ImageNet with images of size $64 \times 64$. In Table 1, we can see that the quantities $\Delta_x$ and $\Delta_{xy}$ do not vary much among the datasets and that the $\Delta$ associated with our our Assump. 1 is two orders of magnitude smaller than the $\Delta$ associated with Saxe et al. [2018]'s assumption indicating the relevance of our assumption.

## 4.2 Linear Autoencoder

For an auto-encoder, we have, $\boldsymbol{Y} = \boldsymbol{X}$. We want to compare the reconstruction properties of $\boldsymbol{W}^{(t)}$ computed though (21) and of the matrix product $\boldsymbol{W}_1^{(t)} \boldsymbol{W}_2^{(t)}$ where $\boldsymbol{W}_1^{(t)}$ and $\boldsymbol{W}_2^{(t)}$ are computed though (22). In this experiment, we have $p = d = 20, n = 1000, r = 5$ and we generated synthetic data. First we generate a *fixed* matrix $\boldsymbol{B} \in \mathbb{R}^{d \times r}$ such that, $\boldsymbol{B}_{kl} \sim \mathcal{U}([0, 1]), 1 \leq k, l \leq n$. Then, for $1 \leq i \leq n$, we sample $\boldsymbol{x}_i \sim \boldsymbol{B}\boldsymbol{z}_i + \boldsymbol{\epsilon}_i$ where $\boldsymbol{z}_i \sim \mathcal{N}(\mathbf{0}, \boldsymbol{D} := \mathrm{diag}(4, 2, 1, 1/2, 1/4))$ and $\boldsymbol{\epsilon}_i \sim 10^{-3}\mathcal{N}(\mathbf{0}, \boldsymbol{I}_d)$. In Fig. 2, we plot the trace norm of $\boldsymbol{W}^{(t)}$ and $\boldsymbol{W}_1^{(t)} \boldsymbol{W}_2^{(t)}$ as well as their respective reconstruction errors as a function of $t$ the number of iterations,

$$\|\boldsymbol{W}^{(t)} - \boldsymbol{B}\boldsymbol{D}\boldsymbol{B}^\top\|_2. \tag{28}$$

We can see that the experimental results are very close to the theoretical behavior predicted with the continuous dynamics in Figure 1. Contrary to the dynamics induced by the linear model formulation ($L = 1$), the dynamics induced by the two-layer linear network ($L = 2$) is very close to a step function: each step corresponds to the learning to a new component: They are learned sequentially.

## 5 Discussion

There is a growing body of empirical and theoretical evidence that the implicit regularization induced by gradient methods is key in the training of deep neural networks. Yet, as noted by Zhang et al. [2017], even for linear models, our understanding of the origin of generalization is limited. In this work, we focus on a simple non-convex objective that is parametrized by a two-layer linear network. In the case of linear regression we show that the *discrete* gradient dynamics also visits points that are implicitly regularized solutions of the initial optimization problem. In that sense, in the context of machine learning, applying gradient descent on the overparametrized model of interest, provides a form of implicit regularization: it sequentially learns the hierarchical components of our problem which could help for generalization. Our setting does not pretend to solve generalization in deep neural networks; many majors components of the standard neural network training are omitted such as the non-linearities, large values of $L$ and the stochasticity in the learning procedure (SGD). Nevertheless, it provides useful insights about the source of generalization in deep learning.

**Acknowledgments.**

This research was partially supported by the Canada CIFAR AI Chair Program, the Canada Excellence Research Chair in "Data Science for Realtime Decision-making", by the NSERC Discovery Grant RGPIN-2017-06936, by a graduate Borealis AI fellowship and by a Google Focused Research award.

## Footnotes

[2]By a rescaling of the data, their proof is valid for any matrix $\boldsymbol{\Sigma}_x$ *proportional* to the identity matrix.

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
