[Supplementary Material]

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

where $\mathbf{\Sigma}_{xy} := \frac{1}{n} \mathbf{X}^\top \mathbf{Y}$ and $\mathbf{\Sigma}_x := \frac{1}{n} \mathbf{X}^\top \mathbf{X}$. Thus, $\mathbf{W}(t)$ is the solution of the differential equation,

$$\dot{\mathbf{W}}(t) = \mathbf{\Sigma}_{xy} - \mathbf{\Sigma}_x \mathbf{W}(t), \quad \mathbf{W}(0) = \mathbf{W}_0. \tag{8}$$

**Proposition 1.** *For any $\mathbf{W}_0 \in \mathbb{R}^{d \times p}$, the solution to the linear differential equation (8) is*

$$\mathbf{W}(t) = e^{-t\mathbf{\Sigma}_x}(\mathbf{W}_0 - \mathbf{\Sigma}_x^\dagger \mathbf{\Sigma}_{xy}) + \mathbf{\Sigma}_x^\dagger \mathbf{\Sigma}_{xy}, \tag{9}$$

*where $\mathbf{\Sigma}_x^\dagger$ is the pseudoinverse of $\mathbf{\Sigma}_x$.*

This standard result on ODE is provided in §B.1. Note that when $\mathbf{W}_0 \to \mathbf{0}$ we have

$$\mathbf{W}(t) \underset{\mathbf{W}_0 \to 0}{\to} (\mathbf{I}_d - e^{-t\mathbf{\Sigma}_x})\mathbf{\Sigma}_x^\dagger \mathbf{\Sigma}_{xy}. \tag{10}$$

**Deep linear network:** $L \geq 2$. The study of the deep linear model is more challenging since for $L \geq 2$, the landscape of the objective function $f$ is non-convex. The gradient flow of (2) is

$$\nabla f_{\mathbf{W}_l}(\mathbf{W}_{[L]}) = \mathbf{W}_{1:l-1}^\top(\mathbf{\Sigma}_x \mathbf{W} - \mathbf{\Sigma}_{xy})\mathbf{W}_{l+1:L}^\top \quad \text{where} \quad \mathbf{W}_{i:j} := \mathbf{W}_i \cdots \mathbf{W}_j, \ 1 \leq l \leq L, \tag{11}$$

where we used the convention that $\mathbf{W}_{1,0} = \mathbf{I}_d$ and $\mathbf{W}_{L+1,L} = \mathbf{I}_p$. Thus (6) becomes

$$\dot{\mathbf{W}}_l(t) = \mathbf{W}_{1:l-1}(t)^\top(\mathbf{\Sigma}_{xy} - \mathbf{\Sigma}_x \mathbf{W}(t))\mathbf{W}_{l+1:L}(t)^\top, \quad \mathbf{W}_l(0) \in \mathbb{R}^{d \times p}, \quad 1 \leq l \leq L. \tag{12}$$

We obtain a *coupled* differential equation (12) that is harder to solve than the previous linear differential equation (8) due, at the same time, to its non-linear components and to the coupling between $\mathbf{W}_l$, $1 \leq l \leq L$. However, in the case $L = 2$, Saxe et al. [2018] managed to find an explicit solution to this coupled differential equation under the assumption that "perceptual correlation is minimal" ($\mathbf{\Sigma}_x = \mathbf{I}_d$).[2] In this work we extend Saxe et al. [2018, Eq. 7] (for $L = 2$) under weaker assumptions. More precisely, we do not require the covariance matrix $\mathbf{\Sigma}_x$ to be the identity matrix. Let $(\mathbf{U}, \mathbf{V}, \mathbf{D})$ be the SVD of $\mathbf{\Sigma}_{xy}$, our assumption is the following:

**Assumption 1.** *There exist two orthogonal matrices $\mathbf{U}$, $\mathbf{V}$ such that we have the joint decomposition,*

$$\mathbf{\Sigma}_x = \mathbf{U}(\mathbf{D}_x + \mathbf{B})\mathbf{U}^\top \quad and \quad \mathbf{\Sigma}_{xy} = \mathbf{U}\mathbf{D}_{xy}\mathbf{V}^\top, \tag{13}$$

*where $\mathbf{B}$ is such that $\|\mathbf{B}\|_2 \leq \epsilon$ and $\mathbf{D}_x$, $\mathbf{D}_{xy}$ are matrices only with diagonal coefficients. We note $\sigma_1 \geq \cdots \geq \sigma_{r_{xy}} > 0$ the singular values of $\mathbf{\Sigma}_{xy}$ and $\lambda_1, \ldots, \lambda_{r_x}$ the diagonal entries of $\mathbf{D}_x$.*

Since two matrices commute if and only if they are co-diagonalizable [Horn et al., 1985, Thm. 1.3.21], the quantity $\epsilon$ represent to what extend $\mathbf{\Sigma}_x$ and $\mathbf{\Sigma}_{xy}\mathbf{\Sigma}_{xy}^\top$ do not commute. Before solving (12) under Assump. 1, we describe some motivating examples where the quantity $\epsilon$ is small or zero:

- **Linear autoencoder**: If $\mathbf{Y}$ is set to $\mathbf{X}$ and $L = 2$, we recover a linear autoencoder: $\hat{\mathbf{x}}(\mathbf{x}) = \mathbf{W}_2^\top \mathbf{W}_1^\top \mathbf{x}$, where $\mathbf{h} := \mathbf{W}_1^\top \mathbf{x}$ is the *encoded* representation of $\mathbf{x}$,

$$\mathbf{\Sigma}_{xy}\mathbf{\Sigma}_{xy}^\top = \left(\frac{1}{n}\mathbf{X}^\top \mathbf{X}\right)^2 = \mathbf{\Sigma}_x^2. \qquad \text{Thus, } \mathbf{

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

$$W_{t+1} = W_t - \eta(\Sigma_x W_t - \Sigma_{xy}) = (I_d - \eta\Sigma_x)W_t + \eta\Sigma_{xy} \,.$$

Noticing that for $1/\lambda_{\max}(\Sigma_x) > \eta > 0$, $I_d - \eta\Sigma_x$ is invertible, this recursion (see §B.4) leads to,

$$W_t = (W_0 - \Sigma_x^\dagger \Sigma_{xy})(I_d - \eta\Sigma_x)^t + \Sigma_x^\dagger \Sigma_{xy} \,. \tag{21}$$

We obtain a similar result as the solution of the differential equation given in Prop. 1. With a vanishing initialization we reach a function that does not sequentially learn some components.

**Two-layer linear model.** The discrete update scheme for the *two-layer linear network* (2) is,

$$W_1^{(t+1)} = W_1^{(t)} - \eta(\Sigma_x W^{(t)} - \Sigma_{xy})(W_2^{(t)})^\top \,, \quad W_2^{(t+1)} = W_2^{(t)} - \eta(W_1^{(t)})^\top(\Sigma_x W^{(t)} - \Sigma_{xy}) \,.$$

When $\epsilon = 0$, by a change of basis and a proper initialization, we can reduce the study of this matrix equation to $r$ independant dynamics (see §B.5 for more details), for $1 \le i \le r$,

$$w_i^{(t+1)} = w_i^{(t)} + \eta w_i^{(t)}(\sigma_i - \lambda_i w_i^{(t)} w_i^{(t)}) \,. \tag{22}$$

Thus we can derive a bound on the iterate $w_i^{(t)}$ leading to the following theorem,

**Theorem 3.** *Under the same assumptions as Thm. 1 and $\epsilon = 0$, we have*

$$W_1^{(t)} = U \operatorname{diag}\left(\sqrt{w_1^{(t)}}, \ldots, \sqrt{w_p^{(t)}}\right) Q \quad and \quad W_2^{(t)} = Q^{-1} \operatorname{diag}\left(\sqrt{w_1^{(t)}}, \ldots, \sqrt{w_d^{(t)}}\right)

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

## A    Deep Linear Autoencoder Recovers PCA.

Let us recall that the two-layer linear autoencoder can be formulated as,

$$(\boldsymbol{W}_2^*, \boldsymbol{W}_1^*) \in \underset{\substack{\boldsymbol{W}_2 \in \mathbb{R}^{r \times p} \\ \boldsymbol{W}_1 \in \mathbb{R}^{d \times r}}}{\arg\min} \frac{1}{2n} \|\boldsymbol{X} - \boldsymbol{X}\boldsymbol{W}_1\boldsymbol{W}_2\|_2^2. \tag{29}$$

Thus, the gradients of the objective are,

$$\nabla f_{\boldsymbol{W}_2}(\boldsymbol{W}_2, \boldsymbol{W}_1) = \boldsymbol{W}_1^\top (\boldsymbol{\Sigma}_x \boldsymbol{W}_1 \boldsymbol{W}_2 - \boldsymbol{\Sigma}_x) \quad \text{and} \quad \nabla f_{\boldsymbol{W}_1}(\boldsymbol{W}_2, \boldsymbol{W}_1) = (\boldsymbol{\Sigma}_x \boldsymbol{W}_1 \boldsymbol{W}_2 - \boldsymbol{\Sigma}_x) \boldsymbol{W}_2^\top.$$

Thus, if $\boldsymbol{W}_2^{(0)} = (\boldsymbol{W}_1^{(0)})^\top$ and if $[\boldsymbol{W}_1^{(0)} \boldsymbol{W}_2^{(0)}, \boldsymbol{\Sigma}_x] = 0$, we have that,

$$\nabla f_{\boldsymbol{M}}(\boldsymbol{W}_2^{(0)}, \boldsymbol{W}_1^{(0)}) = (\boldsymbol{W}_1^{(0)})^\top (\boldsymbol{\Sigma}_x \boldsymbol{W}_1^{(0)} \boldsymbol{W}_2^{(0)} - \boldsymbol{\Sigma}_x) \tag{30}$$

$$= ((\boldsymbol{\Sigma}_x \boldsymbol{W}_1^{(0)} \boldsymbol{W}_2^{(0)} - \boldsymbol{\Sigma}_x)(\boldsymbol{W}_2^{(0)})^\top)^\top \tag{31}$$

$$= \nabla f_{\boldsymbol{N}}(\boldsymbol{W}_2^{(0)}, \boldsymbol{W}_1^{(0)})^\top. \tag{32}$$

Thus, for the discrete case, by a recurrence we have that, $\boldsymbol{W}_1^{(t)} = (\boldsymbol{W}_2^{(t)})^\top$, $t \geq 0$ and for the continuous case, invoking the Cauchy-Lipschitz theorem, we have that $\boldsymbol{W}_1(t) = \boldsymbol{W}_2(t)^\top$, $t \geq 0$. Consequently, the limit solution is such that

$$\boldsymbol{W}_1^* \in \underset{\boldsymbol{W}_1 \in \mathbb{R}^{d \times r}}{\arg\min} \frac{1}{2n} \|\boldsymbol{X} - \boldsymbol{X}\boldsymbol{W}_1\boldsymbol{W}_1^\top\|_2^2, \tag{33}$$

which is a formulation of the PCA.

## B    Proof of Theorems and Propositions

### B.1    Proof of Prop. 1

**Proposition' 1.** *For any $\boldsymbol{W}_0 \in \mathbb{R}^{d \times p}$, the solution to the linear differential equation* (8) *is,*

$$\boldsymbol{W}(t) = e^{-t\boldsymbol{\Sigma}_x}(\boldsymbol{W}_0 - \boldsymbol{\Sigma}_x^\dagger \boldsymbol{\Sigma}_{xy}) + \boldsymbol{\Sigma}_x^\dagger \boldsymbol{\Sigma}_{xy}, \tag{34}$$

*where $\boldsymbol{\Sigma}_x^\dagger$ is the pseudoinverse of $\boldsymbol{\Sigma}_x$.*

We can differentiate (34) and check if it verifies (8). In order to do that, we just need to notice that $\boldsymbol{\Sigma}_x \boldsymbol{\Sigma}_x^\dagger \boldsymbol{\Sigma}_{xy} = \boldsymbol{\Sigma}_{xy}$. To see that we compute the SVD of $\boldsymbol{X}^\top = \boldsymbol{U}^\top \boldsymbol{D} \boldsymbol{V}$ where $\boldsymbol{D}$ is a rectangular matrix with only diagonal coefficients such that,

$$\boldsymbol{D}\boldsymbol{D}^\top = \operatorname{diag}(\lambda_1, \ldots, \lambda_r, 0, \ldots, 0). \tag{35}$$

Thus,      we     have      $\boldsymbol{\Sigma}_x$      $=$      $\boldsymbol{U}^\top \operatorname{diag}(\lambda_1, \ldots, \lambda_r, 0, \ldots, 0)\boldsymbol{U}$      and $\boldsymbol{\Sigma}_x^\dagger = \boldsymbol{U}^\top \operatorname{diag}(1/\lambda_1, \ldots, 1/\lambda_r, 0, \ldots, 0)\boldsymbol{U}$. Leading to,

$$\boldsymbol{\Sigma}_x \boldsymbol{\Sigma}_x^\dagger \boldsymbol{\Sigma}_{xy} = \boldsymbol{U}^\top \boldsymbol{D} \boldsymbol{V} \boldsymbol{Y} = \boldsymbol{\Sigma}_{xy}.$$

Consequently, the matrix valued function $\boldsymbol{W}(t)$ defined in (34) verifies (8). Now we just need to use Cauchy-Lipschitz theorem [Coddington and Levinson, 1955] (a.k.a. Picard–Lindelöf theorem) to say that this solution is the unique solution of the ODE (8).

### B.2    proof of Thm. 1

**Commutative case, $\epsilon = 0$:**    We use ideas from [Saxe et al., 2018] and combine it with Assum. 1 for $\epsilon = 0$. Note that $\epsilon = 0$ if and only if $\boldsymbol{\Sigma}_x$ and $\boldsymbol{\Sigma}_{xy}$ commute. thus, we have that,

$$\boldsymbol{\Sigma}_{xy} = \boldsymbol{U}\boldsymbol{D}_{xy}\boldsymbol{V}^\top \quad \text{and} \quad \boldsymbol{\Sigma}_x = \boldsymbol{U}\boldsymbol{D}_x\boldsymbol{U}^\top. \tag{36}$$

Let us consider a generalization of the linear transformation proposed by Saxe et al. [2018, Eq. S6,S7],

$$\bar{\boldsymbol{W}}_1 = \boldsymbol{U}^\top \boldsymbol{W}_1 \boldsymbol{Q}_1, \quad \bar{\boldsymbol{W}}_l = \boldsymbol{Q}_{l-1}^{-1} \boldsymbol{W}_l \boldsymbol{Q}_l, \ 2 \leq l \leq L-1, \quad \text{and} \quad \boldsymbol{W}_L = \boldsymbol{Q}_{L-1}^{-1} \boldsymbol{W}_L \boldsymbol{V}, \tag{37}$$

where $\boldsymbol{Q}_l$, $1 \le l \le L-1$ are arbitrary invertible matrices. Then, noting $\boldsymbol{Q}_0 := \boldsymbol{U}$ and $\boldsymbol{Q}_L := \boldsymbol{V}$, we get the following dynamics,

$$\frac{d\bar{\boldsymbol{W}}_l(t)}{dt} = \boldsymbol{Q}_{l-1}^{-1}\boldsymbol{W}_{1:l-1}(t)^\top(\boldsymbol{\Sigma}_{xy} - \boldsymbol{\Sigma}_x\boldsymbol{W}(t))\boldsymbol{W}_{l+1:L}(t)^\top\boldsymbol{Q}_l, \quad 1 \le l \le L. \tag{38}$$

Thus using (36), the fact that $\boldsymbol{U}^\top\boldsymbol{U} = \boldsymbol{I}_d$ and that for any invertible matrix $\boldsymbol{Q}$, we have $(\boldsymbol{Q}^{-1})^\top = (\boldsymbol{Q}^\top)^{-1}$, we get that,

$$\frac{d\bar{\boldsymbol{W}}_l(t)}{dt} = \bar{\boldsymbol{W}}_{1:l-1}(t)^\top(\boldsymbol{D}_{xy} - \boldsymbol{D}_x\bar{\boldsymbol{W}}(t))\bar{\boldsymbol{W}}_{l+1:L}(t)^\top, \quad \boldsymbol{W}_l(0) = \boldsymbol{W}_l^{(0)} \ 1 \le l \le L. \tag{39}$$

Using the same argument as [Saxe et al., 2018], if $\bar{\boldsymbol{W}}_l(t)$, $1 \le l \le L$ only have diagonal coefficients then their derivative also only have diagonal coefficients. Thus, if we initialize $\boldsymbol{W}_l^{(0)}$, $1 \le l \le L$, only with diagonal coefficients we have a decoupled solution for each diagonal coefficient. This argument can be formalized using Cauchy-Lipschitz theorem: (39) has a unique solution which is the one we will exhibit in the following.

Recall that we noted $r_0 = d$ and $r_L = p$ and that $\boldsymbol{W}_l \in \mathbb{R}^{r \times l-1, r_l}$. Let us note, $r = \min\{r_l : 0 \le l \le L-1\}$ and $w_{l,i}(t)$, $1 \le i \le r$ the respective diagonal coefficients of $\boldsymbol{W}_l(t)$ for $1 \le l \le L$. Note that for $i \ge r$ one could define diagonal coefficients for some of the matrices $\boldsymbol{W}_l$ but their gradient will be equal to 0, thus non-trivial dynamics only occur for $i \le r$. They follow the equation,

$$\dot{w}_{l,i}(t) = w_{-l,i}(t)(\sigma_i - \lambda_i w_i(t)), \ w_{l,i}(0) \in \mathbb{R}, \quad 1 \le l \le L, \quad 1 \le i \le r, \tag{40}$$

where the notation $w_{-l,i}(t)$ stands for the product of the $w_{k,i}(t)$, $1 \le k \le L$ omitting $w_{l,i}(t)$, i.e.,

$$w_{-l,i}(t) := \prod_{\substack{k=1 \\ k \ne l}}^{L} w_{k,i}(t), \tag{41}$$

and $w_i(t)$ stands for the product of the $w_{k,i}(t)$, $1 \le k \le L$. The difference with [Saxe et al., 2018] is that, since they only consider the case $\boldsymbol{\Sigma}_x = \boldsymbol{I}_d$ they have $\lambda_i = 1$, they also only consider the case $L = 2$. The use of Assumption 1 allowed us to work in a more general case.

We will assume that if $w_{l,i}(t) = w_{k,i}(t)$, $1 \le l, k \le L$, to find an analytic solution and then show that if $w_{l,i}(0) = w_{k,i}(0)$, $1 \le k, l \le L$ then this analytic solution verifies (40) and thus, by Cauchy-Lipschitz theorem, is the unique solution of the non-linear differential equation.

Thus, considering $w_i(t) := w_{1,i}(t) \cdots w_{L,i}(t)$, and assuming that $w_{l,i}(t) = w_{k,i}(t)$, $1 \le l, k \le L$, we get that, for $1 \le i \le r$,

$$\dot{w}_i(t) = \sum_{l=1}^{L} w_{1,i}(t) \cdots w_{l-1,i}(t)\dot{w}_{l,i}(t)w_{l+1,i}(t) \cdots w_{L,i}(t) \tag{42}$$

$$= L w_i(t)^{2-2/L}(\sigma_i - \lambda_i w_i(t)), \ w_i(0) \in \mathbb{R}. \tag{43}$$

**Lemma 1.** *If $w_i(0) \in (0, \frac{\sigma_i}{\lambda_i})$, then the differential equations has a unique solution that is increasing and $w_i(t) \in (0, \frac{\sigma_i}{\lambda_i})$, $\forall t \in \mathbb{R}$.*

*Proof.* If at a time $t \in \mathbb{R}$, we have $w_i(t) = 0$ and thus $\dot{w}_i(t)$. Noticing that then the constant function $w_i(t) = 0 \ t \in \mathbb{R}$ is a solution of (1), by Cauchy-Lipschitz it is the only one. We can use the same argument to say that if there exists a time $t \in \mathbb{R}$, such we have $w_i(t) = 0$ then $w_i(t) = 0 \ \forall t \in \mathbb{R}$. Thus by continuity of $w_i(t)$ we have that if $w_i(0) \in (0, \frac{\sigma_i}{\lambda_i})$ then, $w_i(t) \in (0, \frac{\sigma_i}{\lambda_i})$, $\forall t \in \mathbb{R}$. $\square$

**Case $L = 2$:** in that case we have two situations, $\sigma_i > 0$ and $\sigma_i = 0$, $\lambda_i > 0$ (the case $\sigma_i = \lambda_i = 0$ give a constant functions).

For $\sigma_i > 0$ we have that,

$$t = \int_0^t \frac{dw_i(t)}{2w_i(t)(\sigma_i - \lambda_i w_i(t))} \tag{44}$$

$$= \frac{1}{2\sigma_i} \int_0^t \frac{dw_i(t}{w_i(t)} + \frac{\lambda_i dw_i(t}{\sigma_i - \lambda_i w_i(t)} \tag{45}$$

$$= \frac{1}{2\sigma_i} \ln \frac{w_i(t)(\sigma_i - \lambda_i w_i(0))}{w_i(0)(\sigma_i - \lambda_i w_i(t))} \,. \tag{46}$$

Leading to,

$$w_i(t) = \frac{w_i(0)\sigma_i e^{2\sigma_i t}}{w_i(0)\lambda_i(e^{2\sigma_i t} - 1) + \sigma_i} \,. \tag{47}$$

In order to get a solution for $w_{2,i}(t)$ and $w_{1,i}(t)$, we will use Cauchy-Lipschitz theorem [Coddington and Levinson, 1955]. The idea is that if we find a solution of (40), it is the only one. Let us set, $m_i(0) = n_i(0) = e^{-\delta_i}$, then we can set,

$$w_{2,i}(t) = w_{1,i}(t) = \sqrt{\frac{\sigma_i e^{2\sigma_i t - 2\delta_i}}{\lambda_i(e^{2\sigma_i t - 2\delta_i} - e^{-2\delta_i}) + \sigma_i}} \,, \tag{48}$$

and verify that we have,

$$\dot{w}_{2,i}(t) = w_{1,i}(t)(\sigma_i - \lambda_i w_{1,i}(t)w_{2,i}(t)) \,, \ m_i(0) = e^{-\delta_i} \tag{49}$$

$$\dot{w}_{1,i}(t) = w_{2,i}(t)(\sigma_i - \lambda_i w_{1,i}(t)w_{2,i}(t)) \,, \ n_i(0) = e^{-\delta_i} \quad 1 \le i \le r \,. \tag{50}$$

Thus, this is the unique solution of (40).

For $\sigma_i = 0 \,, \ \lambda_i > 0$ we have that,

$$t = \int_0^t \frac{\dot{w}_i(t)}{-2\lambda_i w_i(t)^2} dt = \frac{1}{2\lambda_i} \left( \frac{1}{w_i(t)} - \frac{1}{w_i(0)} \right) \,. \tag{51}$$

Thus,

$$w_i(t) = \frac{w_i(0)}{1 + 2w_i(0)\lambda_i t} \,. \tag{52}$$

Thus, if we initialize with $m_i(0) = n_i(0) = e^{-\delta_i}$ we get,

$$w_{1,i}(t) = w_{2,i}(t) = \frac{e^{-\delta_i}}{\sqrt{1 + 2e^{-\delta_i}\lambda_i t}} \,. \tag{53}$$

**Non commutative case** $\epsilon > 0$. Now, we will consider Assumption 1 with $\epsilon > 0$ and $L = 2$.

First let us proove two lemmas usefull for later,

**Lemma 2.** *The matrix valued function $\boldsymbol{W}(t)$ converge to $\boldsymbol{X}^\dagger \boldsymbol{Y}$ and thus is bounded for $t > 0$.*

*Proof.* Since (6) is a *gradient dynamics*, it only moves in the span of the gradient of $f$ (the explicit expressions of $\nabla f$ is derived in (7) for $L = 1$ and (11) for a general $L$). We use this property to characterize the solution found by these dynamics. We can study each column of the predictors $\boldsymbol{W} := \boldsymbol{W}_1 \cdots \boldsymbol{W}_L$. If we look at the columns of $\nabla_{\boldsymbol{W}_L} f$, they are included in $\boldsymbol{X}^\top$, thus it means that if we initialize the columns of $\boldsymbol{W}_L^{(0)}$ in that span, then the columns of $\boldsymbol{W}$ will belong to that span during the whole learning process,

$$[\boldsymbol{W}]_i \in \text{span}(\nabla_{\boldsymbol{W}_L} f) \subset \text{span}(\boldsymbol{X}^\top), \ 1 \le i \le n \,, \tag{54}$$

where $\boldsymbol{W}$ is $\boldsymbol{W}(t)$. Thus, if the dynamics (6) converge, then they converge to a matrix with the $i^{th}$ column vector being in the intersection,

$$\text{span}(\boldsymbol{X}^\top) \cap \{\boldsymbol{u} : \boldsymbol{X}\boldsymbol{u} = [\boldsymbol{Y}]_i\} = \{\boldsymbol{X}^\dagger [\boldsymbol{Y}]_i\} \,. \tag{55}$$

Finally, we have $\boldsymbol{X}\boldsymbol{W}(t) \to \boldsymbol{Y}$ by definition of the gradient dynamics,

$$\frac{d\|\boldsymbol{Y} - \boldsymbol{X}\boldsymbol{W}_1(t)\boldsymbol{W}_2(t)\|^2}{dt} = -\|\nabla_{\boldsymbol{W}_1} f(\boldsymbol{W}_1(t), \boldsymbol{W}_2(t))\|^2 - \|\nabla_{\boldsymbol{W}_2} f(\boldsymbol{W}_1(t), \boldsymbol{W}_2(t))\|^2 < 0 \tag{56}$$

$\square$

**Lemma 3.** *We have that* $\|\boldsymbol{W}_1(t)\|^2 = O(t)$ *and* $\|\boldsymbol{W}_2(t)\|^2 = O(t)$.

*Proof.* We have that,

$$\frac{d\|\boldsymbol{W}_1(t)\|^2}{dt} = \langle \boldsymbol{W}_1(t), (\boldsymbol{\Sigma}_{xy} - \boldsymbol{\Sigma}_x \boldsymbol{W}(t))\boldsymbol{W}_2(t)^\top \rangle = Tr((\boldsymbol{\Sigma}_{xy} - \boldsymbol{\Sigma}_x \boldsymbol{W}(t))\boldsymbol{W}(t)^\top). \quad (57)$$

Since $\boldsymbol{W}(t)$ is bounded then $\|\boldsymbol{W}_1(t)\|^2 = O(t)$. The same way we have $\|\boldsymbol{W}_2(t)\|^2 = O(t)$ $\qquad\square$

After the same change of basis as in the commutative case The matrices $\bar{\boldsymbol{W}}_l(t)$ follow the differencial equations

$$\frac{d\bar{\boldsymbol{W}}_1(t)}{dt} = \left(\boldsymbol{D}_{xy} - (\boldsymbol{D}_x + \boldsymbol{B})\bar{\boldsymbol{W}}(t)\right)\bar{\boldsymbol{W}}_2(t)^\top, \quad \frac{d\bar{\boldsymbol{W}}_2(t)}{dt} = \bar{\boldsymbol{W}}_1(t)^\top \left(\boldsymbol{D}_{xy} - (\boldsymbol{D}_x + \boldsymbol{B})\bar{\boldsymbol{W}}(t)\right) \quad (58)$$

In order to perform pertrubation analysis we will use a consequence of Grönwall's inequality [Gronwall, 1919].

**Lemma 4.** *Let $\beta$ be a non negative function and $\alpha$ a non decreasing function. Let $u$ be a function defined on an interval $I = [a, \infty)$ such that*

$$u(t) \leq \alpha(t) + \int_a^t \beta(s)u(s)ds, \quad \forall t \in I. \quad (59)$$

*then we have that*

$$u(t) \leq \alpha(t) \exp\left(\int_a^t \beta(s)ds\right), \quad \forall t \in I. \quad (60)$$

*Proof.* The proof can be found for instance in [Berglund, 2001, Lemma 3.1.6] $\qquad\square$

Thus, let us consider $\bar{\boldsymbol{W}}_1(t)$ and $\bar{\boldsymbol{W}}_2(t)$ the solutions of (58) as well as $\bar{\boldsymbol{W}}_1^0(t)$ and $\bar{\boldsymbol{W}}_2^0(t)$ the solutions of the very same differential equation but with $\boldsymbol{B} = 0$. For notational simplicity we will omit the bar on the matrices $\boldsymbol{W}$ in the following. We have that,

$$\boldsymbol{W}_1(t) - \boldsymbol{W}_1^0(t) = \int_0^t [-\boldsymbol{B}\boldsymbol{W}(s)\boldsymbol{W}_2(s) + (\boldsymbol{D}_{xy} - \boldsymbol{D}_x\boldsymbol{W}(s))\boldsymbol{W}_2(s)^\top - (\boldsymbol{D}_{xy} - \boldsymbol{D}_x\boldsymbol{W}^0(s))\boldsymbol{W}_2^0(s)^\top]ds$$

Leading to

$$\|\boldsymbol{W}_1(t) - \boldsymbol{W}_1^0(t)\| \leq \int_0^t \|D_x\|\|\boldsymbol{W}_1(s)\boldsymbol{W}_2(s)\boldsymbol{W}_2(s)^\top - \boldsymbol{W}_1^0(s)\boldsymbol{W}_2^0(s)\boldsymbol{W}_2^0(s)^\top\|ds$$

$$+ \int_0^t \|D_x y\|\|\boldsymbol{W}_2(s) - \boldsymbol{W}_2^0(t)\|ds + \|\boldsymbol{B}\boldsymbol{W}(t)\boldsymbol{W}_2(t)\|$$

In order to upper bound the first integral we will consider the function $F(\boldsymbol{A}, \boldsymbol{B}) := \boldsymbol{A}\boldsymbol{B}\boldsymbol{B}^\top$. This function is Lipschitz on any compact because this function is infinitely differentiable. Thus we have that (omiting the $t$ in the notation),

$$\|\boldsymbol{W}_1\boldsymbol{W}_2\boldsymbol{W}_2^\top - \boldsymbol{W}_1^0\boldsymbol{W}_2^0(\boldsymbol{W}_2^0)^\top\| = t^{3/2}\|\frac{\boldsymbol{W}_1}{\sqrt{t}}\frac{\boldsymbol{W}_2}{\sqrt{t}}\frac{\boldsymbol{W}_2}{\sqrt{t}}^\top - \frac{\boldsymbol{W}_1^0}{\sqrt{t}}\frac{\boldsymbol{W}_2^0}{\sqrt{t}}(\frac{\boldsymbol{W}_2^0}{\sqrt{t}})^\top\| \quad (61)$$

$$= t^{3/2}\|F(\frac{\boldsymbol{W}_1}{\sqrt{t}}, \frac{\boldsymbol{W}_2}{\sqrt{t}}) - F(\frac{\boldsymbol{W}_1^0}{\sqrt{t}}, \frac{\boldsymbol{W}_2^0}{\sqrt{t}})\|$$

$$\leq t^{3/2}L(\|\frac{\boldsymbol{W}_2}{\sqrt{t}} - \frac{\boldsymbol{W}_2^0}{\sqrt{t}}\| + \|\frac{\boldsymbol{W}_1}{\sqrt{t}} - \frac{\boldsymbol{W}_1^0}{\sqrt{t}}\|)$$

$$\leq tL(\|\boldsymbol{W}_2 - \boldsymbol{W}_2^0\| + \|\boldsymbol{W}_1 - \boldsymbol{W}_1^0\|)$$

Using the fact that $\frac{\boldsymbol{W}_1(t)}{\sqrt{t}}, \frac{\boldsymbol{W}_1^0(t)}{\sqrt{t}}, \frac{\boldsymbol{W}_2^0(t)}{\sqrt{t}}$ and $\frac{\boldsymbol{W}_2(t)}{\sqrt{t}}, t \geq 0$ live in a compact set (Lemma 3) and that $F$ is Lipschitz on any compact. Thus, using that $\|S\| = O(\epsilon)$, we have

$$\|\boldsymbol{W}_1(t) - \boldsymbol{W}_1^0(t)\| \leq O(\epsilon) + O(1)\int_0^t s(\|\boldsymbol{W}_2(s) - \boldsymbol{W}_2^0(s)\| + \|\boldsymbol{W}_1(s) - \boldsymbol{W}_1(s)^0\|)ds$$

The same way we can prove that,

$$\|\boldsymbol{W}_2(t) - \boldsymbol{W}_2^0(t)\| \leq O(\epsilon) + O(1)\int_0^t s(\|\boldsymbol{W}_2(t) - \boldsymbol{W}_2^0(t)\| + \|\boldsymbol{W}_1(t) - \boldsymbol{W}_1^0(t)\|)dt$$

And consequently we can sum these two inequalities and apply Grönwall's inequality with the quantities $u(t) = \|\boldsymbol{W}_1(t) - \boldsymbol{W}_1^0(t)\| + \|\boldsymbol{W}_2(t) - \boldsymbol{W}_2^0(t)\|$, $\alpha(t) = O(\epsilon)$ and $\beta(s) = O(s)$ to get,

$$\|\boldsymbol{W}_1(t) - \boldsymbol{W}_1^0(t)\| + \|\boldsymbol{W}_2(t) - \boldsymbol{W}_2^0(t)\| \leq \epsilon \cdot e^{O(t^2)} \tag{62}$$

### B.3 Proof of Thm. 2

**Theorem' 2.** *Let us denotes $w_i(t)$, the values defined in* (16). *If $m_i(0) = e^{-\delta}$, $1 \leq i \leq r$, then we have,*

$$w_i(\delta t) \xrightarrow[\delta\to\infty]{} \begin{cases} 0 & if \quad t < 1/\sigma_i \\ \sqrt{\frac{\sigma_i}{\lambda_i + \sigma_i}} & if \quad t = 1/\sigma_i \\ \sqrt{\frac{\sigma_i}{\lambda_i}} & otherwise\,. \end{cases}$$

*Proof.* Using (16) we get that,

$$w_i(\delta t) = \sqrt{\frac{\sigma_i e^{2\delta(\sigma_i t - 1)}}{\lambda_i(e^{2\delta(\sigma_i t - 1)} - e^{-2\delta}) + \sigma_i}}\,. \tag{63}$$

Then we can conclude saying that for any $i$ and $t \geq 0$,

$$e^{2\delta(\sigma_i t - 1)} \xrightarrow[\delta\to\infty]{} \begin{cases} 0 & if \quad t < 1/\sigma_i \\ 1 & if \quad t = 1/\sigma_i \\ +\infty & otherwise\,, \end{cases} \tag{64}$$

and that when $\delta \to \infty$,

$$\|\boldsymbol{W}_i^0(\delta t) - \boldsymbol{W}_i^\epsilon(\delta t)\| \leq \epsilon \cdot e^{ct^2} = e^{\delta^2(ct^2 - \ln(\delta))} \to 0 \tag{65}$$

$\square$

### B.4 Proof of Eq. 21

Let us recall (21),

$$\boldsymbol{W}_t = (\boldsymbol{W}_0 - \boldsymbol{\Sigma}_x^\dagger \boldsymbol{\Sigma}_{xy})(\boldsymbol{I}_d - \eta\boldsymbol{\Sigma}_x)^t + \boldsymbol{\Sigma}_x^\dagger \boldsymbol{\Sigma}_{xy}\,. \tag{21}$$

Thus we have that,

$$\boldsymbol{W}_t = \boldsymbol{W}_0(\boldsymbol{I}_d - \eta\boldsymbol{\Sigma}_x)^t + \eta\boldsymbol{\Sigma}_{xy}\sum_{s=0}^{t-1}(\boldsymbol{I}_d - \eta\boldsymbol{\Sigma}_x)^s \tag{66}$$

$$= \boldsymbol{W}_0(\boldsymbol{I}_d - \eta\boldsymbol{\Sigma}_x)^t + \boldsymbol{\Sigma}_x^\dagger \boldsymbol{\Sigma}_{xy} - \boldsymbol{\Sigma}_x^\dagger \boldsymbol{\Sigma}_{xy}(\boldsymbol{I}_d - \eta\boldsymbol{\Sigma}_x)^t$$

$$= (\boldsymbol{W}_0 - \boldsymbol{\Sigma}_x^\dagger \boldsymbol{\Sigma}_{xy})(\boldsymbol{I}_d - \eta\boldsymbol{\Sigma}_x)^t + \boldsymbol{\Sigma}_x^\dagger \boldsymbol{\Sigma}_{xy}\,. \tag{67}$$

### B.5 Proof of Thm. 3

**Case $\epsilon = 0$.** If we define $\boldsymbol{W}^{(t)} := \boldsymbol{W}_1^{(t)}\boldsymbol{W}_2^{(t)}$, the discrete update scheme for the *two-layer linear neural network* (2) is,

$$\begin{cases} \boldsymbol{W}_1^{(t+1)} = \boldsymbol{W}_1^{(t)} - \eta(\boldsymbol{\Sigma}_x\boldsymbol{W}^{(t)} - \boldsymbol{\Sigma}_{xy})(\boldsymbol{W}_2^{(t)})^\top \\ \boldsymbol{W}_2^{(t+1)} = \boldsymbol{W}_2^{(t)} - \eta(\boldsymbol{W}_1^{(t)})^\top(\boldsymbol{\Sigma}_x\boldsymbol{W}^{(t)} - \boldsymbol{\Sigma}_{xy})\,. \end{cases} \tag{68}$$

Using the same transformation (37) as in §B.2 we get that,

$$\begin{cases} \bar{\boldsymbol{W}}_1^{(t+1)} = \bar{\boldsymbol{W}}_1^{(t)} - \eta(\boldsymbol{D}\bar{\boldsymbol{W}}^{(t)} - \boldsymbol{S})(\bar{\boldsymbol{W}}_2^{(t)})^\top \\ \bar{\boldsymbol{W}}_2^{(t+1)} = \bar{\boldsymbol{W}}_2^{(t)} - \eta(\bar{\boldsymbol{W}}_1^{(t)})^\top(\boldsymbol{D}\bar{\boldsymbol{W}}^{(t)} - \boldsymbol{S})\,. \end{cases} \tag{69}$$

where $D$ and $S$ only have diagonal coefficients. Thus, by an immediate recurrence we can show that if $\bar{W}_1^{(0)}$ and $\bar{W}_2^{(0)}$ only have diagonal coefficients then $\bar{W}_1^{(t)}$ and $\bar{W}_2^{(t)}$ for $t \in \mathbb{N}$ only have diagonal coefficients.

let us note, $r = \min(p, d)$ and $m_1^{(t)}, \ldots, m_r^{(t)}$ and $n_1^{(t)}, \ldots, n_r^{(t)}$ the respective diagonal coefficients of $\bar{W}_1^{(t)}$ and $\bar{W}_2^{(t)}$, they follow the equation,

$$m_i^{(t+1)} = m_i^{(t)} + \eta n_i^{(t)} (\sigma_i - \lambda_i n_i^{(t)} m_i^{(t)}) \quad \text{and} \quad n_i^{(t+1)} = n_i^{(t)} + \eta m_i^{(t)} (\sigma_i - \lambda_i n_i^{(t)} m_i^{(t)}), \quad 1 \le i \le r. \tag{70}$$

In order to prove Thm. 3, we will prove several properties on the sequences $(m_i^{(t)})_{t \ge 0}$ and $(n_i^{(t)})_{t \ge 0}$, $1 \le i \le d$. First let us introduce the sequence $(a_i^{(t)})_{t \ge 0}$ defined as $a_i^{(t)} := m_i^{(t)} n_i^{(t)}, t \ge 0$.

**Lemma 5.** *If $m_i^{(0)} = n_i^{(0)}$ then,*

$$m_i^{(t)} = n_i^{(t)} = \sqrt{a_i^{(t)}} \qquad \forall t \in \mathbb{N}. \tag{71}$$

*Proof.* By a straightforward recurrence we have that if at time $t$, $m_i^{(t)} = n_i^{(t)}$ then by (70), we have $m_i^{(t+1)} = n_i^{(t+1)}$. $\qquad \square$

Thus, we will now focus on the sequence $(a_i^{(t)})_{t \ge 0}$, by (70), we have that

$$a_i^{(t+1)} = a_i^{(t)} + 2\eta a_i^{(t)} (\sigma_i - \lambda_i a_i^{(t)}) + \eta^2 a_i^{(t)} (\sigma_i - \lambda_i a_i^{(t)})^2 = a_i^{(t)} + \eta a_i^{(t)} (\sigma_i - \lambda_i a_i^{(t)}) (2 + \eta(\sigma_i - \lambda_i a_i^{(t)})). \tag{72}$$

Similarly as for the continuous case, there is two different behavior $\sigma_i > 0$ ad $\sigma_i = 0$, $\lambda_i > 0$. In the following we assume that $\eta > 0$.

For $\sigma_i > 0$ we can derive the following results,

**Lemma 6.** *For any $1 \le i \le r_{xy}$, if $0 < a_i^{(0)} < \frac{\sigma_i}{\lambda_i}$ and $2\eta\sigma_i < 1$, then, the sequence $(a_i^{(t)})$ is increasing and*

$$0 < a_i^{(t)} < \frac{\sigma_i}{\lambda_i}, \quad \forall t \ge 0. \tag{73}$$

*Proof.* By assumption (73) is true for $t = 0$.

Let us assume that (73) is true for a time-step $t$ and let us prove that it is still true at time-step $t + 1$.

Using the recursive definition (72) of $a_i^{(t)}$ we get for $t \ge 0$,

$$a_i^{(t+1)} = a_i^{(t)} + \eta a_i^{(t)} (\sigma_i - \lambda_i a_i^{(t)})(2 + \eta(\sigma_i - \lambda_i a_i^{(t)})) \tag{74}$$

$$> a_i^{(t)} > 0. \tag{75}$$

For the upper bound we need to notice that $a_i^{(t+1)} = f_i(a_i^{(t)})$ where $f_i : x \mapsto x + \eta x (\sigma_i - \lambda_i x)(2 + \eta(\sigma_i - \lambda_i x))$ where $\eta > 0$. Since we assumed that $2\eta\sigma_i < 1$, we have that,

$$f_i(x) < x + \eta x (\sigma_i - \lambda_i x)(2 + \frac{\sigma_i - \lambda_i x}{2\sigma_i}), \quad \forall x \in (0, 1) \tag{76}$$

$$< x + \frac{x(1 - \frac{\lambda_i}{\sigma_i}x)(\frac{5}{2} - \frac{\lambda_i}{2\sigma_i}x)}{2} =: g_i(x), \quad \forall x \in (0, 1). \tag{77}$$

$$\tag{78}$$

Then we just need to show that $g(x) < \frac{\lambda_i}{\sigma_i}$, $\forall x \in (0, \frac{\lambda_i}{\sigma_i})$.

$$4g'(x) = 9 - 12\frac{\lambda_i}{\sigma_i}x + 3\frac{\lambda_i^2}{\sigma_i^2}x^2 > 0, \quad \forall x \in (0, \frac{\lambda_i}{\sigma_i}). \tag{79}$$

Thus $g$ is non-decreasing on $(0, 1)$ and consequently, $g(x) < g(\frac{\lambda_i}{\sigma_i}) = \frac{\lambda_i}{\sigma_i}$, $\forall x \in (0, 1)$.

Finally, we get that,

$$a_i^{(t+1)} = f_i(a_i^{(t)}) < g(a_i^{(t)}) < g(\tfrac{\lambda_i}{\sigma_i}) = \frac{\lambda_i}{\sigma_i}. \tag{80}$$

$\square$

With this lemma we can proof Thm. 3. Let us first recall this theorem.

**Theorem' 3.** *For any $1 \leq i \leq r_{xy}$, if $\frac{\sigma_i}{\lambda_i} > a_i^{(0)} > 0$ and $2\eta\sigma_i < 1$, then $\forall t \geq 0$, $1 \leq i \leq r$ we have,*

$$a_i^{(t)} \leq \frac{a_i^{(0)}}{(\sigma_i - \lambda_i a_i^{(0)})e^{(-2\eta\sigma_i - \eta^2\sigma_i^2)t} + a_i^{(0)}\lambda_i} \tag{81}$$

$$and \qquad a_i^{(t)} \geq \frac{a_i^{(0)}}{(\sigma_i - \lambda_i a_i^{(0)})e^{(-2\eta\sigma_i + 4\eta^2\sigma_i^2)t} + a_i^{(0)}\lambda_i}, \tag{82}$$

*and for $r_{xy} \leq i \leq r_x$,*

$$a_i^{(t)} \leq \frac{a_i^{(0)}}{1 + a_i^{(0)}\lambda_i \eta t}.$$

*Proof.* In this proof for notational compactness we will remove the index $i$.

We first prove (81), we work with $1/a^{(t+1)} - \frac{\lambda}{\sigma}$, Using (72) we get,

$$1/a^{(t+1)} - \frac{\lambda}{\sigma} = \frac{1}{a^{(t)}}\left(\frac{1}{1 + 2\eta\sigma(1 - \frac{\lambda}{\sigma}a^{(t)}) + \eta^2\sigma^2(1 - \frac{\lambda}{\sigma}a^{(t)})^2}\right) - \frac{\lambda}{\sigma} \tag{83}$$

$$\geq \frac{1}{a^{(t)}}\left(\frac{1}{1 + (2\eta\sigma + \eta^2\sigma^2)(1 - \frac{\lambda}{\sigma}a^{(t)})}\right) - \frac{\lambda}{\sigma} \tag{84}$$

$$\geq \frac{1}{a^{(t)}} - \frac{\lambda}{\sigma} - \frac{2\eta\sigma + \eta^2\sigma^2}{a^{(t)}}(1 - \frac{\lambda}{\sigma}a^{(t)}), \tag{85}$$

where we used that $\frac{1}{1+x} \geq 1 - x$, $\forall x \geq 0$. Thus we have,

$$1/a^{(t)} - \frac{\lambda}{\sigma} \geq (\frac{1}{a^{(t-1)}} - \frac{\lambda}{\sigma})(1 - 2\eta\sigma - \eta^2\sigma^2) \tag{86}$$

$$\geq (\frac{1}{a^{(0)}} - \frac{\lambda}{\sigma})(1 - 2\eta\sigma - \eta^2\sigma^2)^t \tag{87}$$

$$\geq (\frac{1}{a^{(0)}} - \frac{\lambda}{\sigma})e^{(-2\eta\sigma - \eta^2\sigma^2)t}. \tag{88}$$

Thus Leads to,

$$a^{(t)} \leq \frac{\sigma a^{(0)}}{(\sigma - \lambda a^{(0)})e^{t(-2\eta\sigma - \eta^2\sigma^2)} + a^{(0)}\lambda}. \tag{89}$$

To prove (81) we will once again work with $1/a^{(t+1)} - \frac{\lambda}{\sigma}$. Using (72) we get

$$1/a^{(t+1)} - \frac{\lambda}{\sigma} = \frac{1}{a^{(t)}}\left(\frac{1}{1 + 2\sigma(1 - \frac{\lambda}{\sigma}a^{(t)}) + \sigma^2(1 - \frac{\lambda}{\sigma}a^{(t)})^2}\right) - \frac{\lambda}{\sigma} \tag{90}$$

$$\leq \frac{1}{a^{(t)}}\left(\frac{1}{1 + 2\sigma(1 - \frac{\lambda}{\sigma}a^{(t)})}\right) - \frac{\lambda}{\sigma} \tag{91}$$

$$\leq \frac{1}{a^{(t)}} - \frac{\lambda}{\sigma} - \frac{2\eta\sigma}{a^{(t)}}(1 - \frac{\lambda}{\sigma}a^{(t)}) + \frac{4\eta^2\sigma^2}{a^{(t)}}(1 - \frac{\lambda}{\sigma}a^{(t)})^2, \tag{92}$$

where we used that $\frac{1}{1+x} \leq 1 - x + x^2$ , $\forall x \geq 0$. Thus we have,

$$1/a^{(t)} - \frac{\lambda}{\sigma} \leq (\frac{1}{a^{(t)}} - \frac{\lambda}{\sigma})(1 - 2\sigma\eta + 4\eta^2\sigma^2) \tag{93}$$

$$\leq (\frac{1}{a^{(0)}} - \frac{\lambda}{\sigma})(1 - 2\sigma\eta + 4\eta^2\sigma^2)^t \tag{94}$$

$$\leq (\frac{1}{a^{(0)}} - \frac{\lambda}{\sigma})e^{(-2\sigma\eta + 4\eta^2\sigma^2)t} \,. \tag{95}$$

This leads to,

$$a^{(t)} \geq \frac{\sigma a^{(0)}}{(\sigma - \lambda a^{(0)})e^{t(-2\sigma + 4\eta^2\sigma^2)} + \lambda a^{(0)}} \,. \tag{96}$$

Now for $\sigma = 0$ and $\lambda > 0$ we have that,

$$a_i^{(t+1)} = a_i^{(t)}(1 - 2\lambda\eta a_i^{(t)} + \lambda^2\eta^2(a_i^{(t)})^2) \,. \tag{97}$$

Thus, considering $(a_i^{(t)})^{-1}$ we get

$$1/a_i^{(t+1)} = 1/a_i^{(t)}(1 - 2\lambda\eta a_i^{(t)} + \lambda^2\eta^2(a_i^{(t)})^2)^{-1} \tag{98}$$

$$\geq 1/a_i^{(t)}(1 + 2\lambda\eta a_i^{(t)} - \lambda^2\eta^2(a_i^{(t)})^2) \tag{99}$$

$$= 1/a_i^{(t)} + 2\lambda\eta - \lambda^2\eta^2 a_i^{(t)} \,. \tag{100}$$

Thus, if we assume that $1/a_i^{(0)} \geq \lambda\eta$ we have that $(1/a_i^{(t)})$ is a increasing sequence and that,

$$1/a_i^{(t)} \geq 1/a_i^{(t-1)} + \lambda\eta \geq 1/a_i^{(0)} + \lambda\eta t \,, \tag{101}$$

leading to,

$$a_i^{(t)} \leq \frac{a_i^{(0)}}{1 + a_i^{(0)}\lambda\eta t} \,. \tag{102}$$

$\square$

**Case $\epsilon > 0$.** If we are able to show that all the sequences $W_1^{(t)}$ and $W_2^{(t)}$ are bounded

From this theorem we can deduce the following corollary,

**Proof of Corollary 3** Let us recall Corollary 3.

**Corollary' 3.** If $\eta < \frac{1}{2\sigma_1}$, $\eta < 2\frac{\sigma_i - \sigma_{i+1}}{\sigma_i^2}$ and $\eta < \frac{\sigma_i - \sigma_{i+1}}{2\sigma_{i+1}^2}$ , $\forall i \ 1 \leq i \leq r_{xy} - 1$ then for $1 \leq i < r_x$,

$$a_i^{(\delta T_j)} \underset{\delta\to\infty}{\to} \begin{cases} 0 & \text{if} \ \ i > r_{xy} \ \ \text{or} \ \ j < i \\ \frac{\sigma_i}{\lambda_i} & \text{if} \ \ i \leq r_{xy} \ \ \text{and} \ \ j > i \,, \end{cases}$$

where $T_j := \frac{1}{\sigma_j\eta}$ , $1 \leq j \leq r_{xy}$ and $T_j = +\infty$ if $j > r_{xy}$ and $T_0 = 0$.

*Proof.* First let us notice that since $\sigma_1 > \ldots > \sigma_{r_{xy}} > 0$, the assumption $\eta < 1/(2\sigma_1)$ implies $\eta < 1/(2\sigma_i)$ , $1 \leq i \leq d$.

Let $i \leq r_{xy}$. Let us first prove that if $j < i$, then $a_i^{(T_j)} \underset{\delta\to\infty}{\to} 0$.

Using (81) and recalling that in Thm. 3, we assume that $a_i^{(0)} = e^{-2\delta}$, we have for $1 \leq j < i$,

$$0 < a_i^{(\delta T_j)} \leq a_i^{(\delta T_{i-1})} < \frac{\sigma_i}{(\sigma_i e^{2\delta} - \lambda_i)e^{\delta(-2\sigma_i/\sigma_{i-1} - \eta\sigma_i^2/\sigma_{i-1})} + \sigma_i} \underset{\delta\to\infty}{\to} 0 \,. \tag{103}$$

We have $(2 + \eta\sigma_i)\sigma_i/\sigma_{i-1} < 2$, because we assumed that $\eta < 2\frac{\sigma_i - \sigma_{i+1}}{\sigma_i^2}$ , $\forall i \ \ 1 \leq i \leq d$. Note that for $i = r_{xy}$ we have $a_i^{(\delta T_{i+1})} = 0$ , $\forall \delta > 0$.

Let us now prove that if $j > i$, then $a_i^{(\delta T_j)} \underset{\delta \to \infty}{\to} \frac{\sigma_i}{\lambda_i}$.

Using (82) and recalling that in Thm. 3, we assume that $a_i^{(0)} = e^{-2\delta}$, we have for $j > i \geq 1$,

$$\frac{\lambda_i}{\sigma_i} > a_i^{(\delta T_j)} \geq a_i^{(\delta T_{i+1})} > \frac{\sigma_i}{(\sigma_i e^{2\delta} - \lambda_i)e^{\delta(-2\sigma_i/\sigma_{i+1} + 4\eta\sigma_i^2/\sigma_{i+1})} + \sigma_i} \underset{\delta \to \infty}{\to} \frac{\sigma_i}{\lambda_i}. \tag{104}$$

where we have that $(e^{2\delta} - 1)e^{\delta(-2\sigma_i/\sigma_j + 4\eta\sigma_i^2/\sigma_j)} \to 0$ because

$$1 - \sigma_i/\sigma_{i+1} + 2\eta\sigma_i^2/\sigma_{i+1} < 0 \Leftrightarrow \eta < (\sigma_i - \sigma_{i+1})/(2\sigma_i^2).$$

Now for $i \geq r_{xy} + 1$ we just need to use, (83) to get,

$$a_i^{(t)} \leq \frac{a_i^{(0)}}{1 + \lambda\eta a_i^{(0)} t} \underset{\delta \to \infty}{\to} 0. \tag{105}$$