[Reviews · NeurIPS 2019]

Reviewer 1



The authors theoretically analyze the dynamics of continuous and discrete optimization for one and two layer linear neural networks under square loss. For the continuous case, they improve the results of Saxe et al. by proving the implicit regularization under more reasonable assumptions. They also analyze the behavior of discrete dynamics for the same networks. To the best of my knowledge, analysis of discrete dynamics is novel. In theorem 1, I understand that the specific initialization is required in your proof for the results to hold. I wonder to what extent this initialization is necessary? If one initializes the matrices to a matrix with vanishing nuclear norm but different singular vectors, I guess we still should observe the same phenomenon of sequential learning. Can the authors clarify to what extent this initialization form is necessary? The fact that the authors were able to relax the assumption that the two matrices should commute is very interesting. A question similar to the previous one regarding this assumption: is this assumption that the two matrices should approximately commute just needed for the proof or do the authors think it is necessary?

Reviewer 2



The paper studies the gradient dynamics of the SGD learning algorithm in the context of the two layer linear network. The authors consider two scenarios: continuous and discrete gradient dynamics. The former has been studied in Saxe et al 2018 but the authors manage to loosen the assumption and provide a more general result. In particular, under the assumption that both the covariance matrix X' X and correlation matrix X' Y share the join decomposition, SGD updates can be derived to have a close form. Additionally, the paper shows an interesting asymptotic phenomenon: as the initialization of weights tends to go zero, the weight product at any SGD iterate converges to the solution of the reduced rank regression problem. A more novel result is the formulation of the discrete gradient dynamics where the authors show that under a bit more stringent assumption, the SGD updates can have a close form. Although the paper shows interesting and novel theoretical results, the main concern is that the assumption 1 is too strong. It is not clear that this assumption is met in practice. Furthermore, the theoretical analysis might not be able to generalize to network of more than 3 layers. For this, the paper is more suitable to be considered as a low rank regression problem, rather than a deep linear neural network.

Reviewer 3



Summary: This paper studies the implicit regularization of discrete gradient descent algorithm for over-parameterized two-layer neural networks. By making mild assumptions on the data, this paper finds out that the discrete gradient dynamics learn the reduced-rank regression solutions sequentially. The experimental results support the theoretical results in this paper. Pros: - The theoretical results, i.e., gradient dynamics sequentially learn the solutions, on the reduced-rank regression problem are novel and interesting. The results can shed lights on representation learning of deep neural networks. - The experiments on both synthetic and real datasets are well-designed, which support the theoretical results as well as validate the assumptions. Limitation & Questions: - The analysis seems to be specific to two-layer linear neural networks. Could this be possibly extended to deep neural networks? Typo: - L459-L460, \bm{W}_{1}(t)^{0}. Other Comments: The code 'M_0 = 10**-3 * 1/np.sqrt(d) * np.random.rand(r,d)' in [18], 'r' is not defined. After replacing r with a number, the code can reproduce the results in Figure 2. There is an omission in the related work on implicit regularization: https://arxiv.org/abs/1712.09203

Reviewer 4



The paper addresses an important topic (implicit regularization in deep learning), is well-written, and although I did not verify proofs in the appendixes, I believe it is mathematically solid. Nonetheless, I have several concerns with regards to originality, significance and clarity (see below), which ultimately lead me to vote against its acceptance. *Originality:* The setup and result proven here are very similar to those of previous works (in particular [2], which was not cited!). This is true for the proof techniques as well. The only part I view as potentially novel is the perturbation analysis, but that is unfortunately not discussed at all in the body of the paper. I recommend to the authors to put much more focus on this aspect, as without it the paper is merely a straightforward extension of prior work. *Significance:* In my opinion, the significance of contribution (A) --- relaxation of assumption in (i) --- is in the perturbation analysis, i.e. the fact that only approximate alignment is required. Contribution (B) --- transition from continuous (gradient flow) to discrete (gradient descent) optimization --- is also relatively simple without the perturbation analysis. Since that is fully deferred to the appendixes, I think the paper body itself falls short of the significance threshold required for publication at NeurIPS. *Clarity:* The paper is in my opinion clear and well-written. However, I would like to point out that its presentation may potentially mislead readers into overestimating the significance and implications of the results proven. Specifically, the paper repeatedly uses the term "deep" (including in its title), where in fact its analyses are all limited to shallow (two or one layer) models. Also, it refers to "implicit regularization" (including in the title), and talks about gradient descent over different architectures biasing towards certain solutions, but the setting it treats is one in which the final solution is known, and the only implication of the architecture is on the path taken to get there. I am not claiming that the latter question is uninteresting, but it should be contrasted against the kind of implicit regularization we typically encounter in deep learning (early stopping is not needed there). I am aware of the fact that there are prior works that make the same inaccuracies, but in my opinion that does not justify doing so. *Quality:* I did not go over proofs in the appendixes, but as far as I could tell the paper is technically solid. *Additional Comments:* * In Figure 2 (experiment), I believe there should be a discussion on the one layer model overfitting after some point and the two layer model not doing so. * It is not clear to me if the discrete result allows epsilon > 0 (i.e. approximate alignment). The text suggests that it does but the statement doesn't appear to deliver. If there is no accounting for approximation in the discrete case then my concern about significance is even more potent. -------- Update after reading other reviews and author response: In high level, my critique on this paper is that it is very similar to prior work (in particular Lampinen and Ganguli 2019, which was not cited) in its formulation and assumptions (and also in many of its proof techniques), but nonetheless does not clearly articulate its distinction. I believe an uninformed reader can easily misinterpret the text into associating with this work contributions that have already been made. A more transparent presentation in my view would be to start off with a detailed technical recap of the existing framework on which this paper relies, and then move on to discuss the incremental improvements this paper introduces. Since the improvements are ultimately incremental, I think their technical aspects (e.g. proof ideas) must be included in the body of the paper, not just their motivation. For example, it is important to clarify that the discrete result does not include perturbations, thus the assumptions ensure that the multidimensional problem decomposes into scalar problems, immensely simplifying the analysis. The author response did not really address my concerns. For example, while I recommend that technical details behind perturbation and discrete analyses be added to paper body, the response refers to text discussing motivation and challenges. The comment I gave on misuse of the term "implicit regularization" was also not recognized --- in practical deep leaning the term refers to the final solution being regularized, not just the optimization path (again, I am not saying paths are not interesting, just that the text should be transparent). After reading the author response, I reviewed the proofs in the appendix. I believe they do carry some novelty, and if added to paper body, would bring forth a contribution worthy of acceptance. I encourage the authors to rewrite the paper in a way that is more transparent on one hand, and on the other does more justice to their technical novelty. With regards to the current form of the paper, I think it is marginally below acceptance threshold (updating my score accordingly).

[Author Response · NeurIPS 2019]

First we would like to thank the reviewers for their interest on the contributions of the main paper. We share the enthusiasm of the reviewers about the promising theoretical results on the discrete dynamics and the perturbation analysis of the paper and we highly appreciated their interest and their detailed comments.

**Regarding the strength of Assumption 1. (R1 & R3 & R4)** Eq. 13 is always true using a large enough $\epsilon$ (one can always find such decomposition described in Eq. 13), but the matrix $B$ may be large (L 279 we describe how to compute a candidate for $B$ in this decomposition). Formally speaking, Assumption 1 should be stated as a proposition and the informal assumption associated with this proposition is that the matrix $B$ (or equivalently epsilon) is relatively small. As noted by reviewer 1 and reviewer 4 this assumption is significantly weaker than the one done in the related work and thus is a major improvement compare to them. Experiments in section 4.1 assesses that this assumption is actually quite met in practice. We thank thanks the reviewers for pointing this out, we will clarify this in the revision.

**Results for no more than two layers. (R2 & R3 & R4)** We agree with the reviewers that we should remove "deep" from the title. On the question whether or not we can extend our analysis to more than two layers, it is a very interesting question. To our knowledge, proving such result is still an open question, the reason being that there is not closed form solutions for the continuous dynamics when $n \geq 3$ (previous related works used proof technique necessitating closed form solutions of the continuous dynamics). However, our discrete analysis and our perturbation analysis did not use a closed form solution, letting us thinks that we can be optimistic and that we could use similar techniques for $n \geq 3$.

**"the perturbation analysis, [...] not discussed at all in the body of the paper". (R4)** We think that R4 missed how we address perturbation analysis in the body of the paper: we discuss the practical relevance of the assumption $\epsilon$ small (L134-149) providing several application cases and provide intuitions regarding this hypothesis: $\epsilon$ represent to what extent the covariance matrices $\Sigma_x$ and $\Sigma_{xy}$ do not commute (L133).

We provide experimental evidence of the relevance of this assumption ($\epsilon$ small) in §4.1. We consider that the motivations behind the perturbation analysis are well discussed as noted by R1 and R3. We decided not to discuss the details of the proof technique itself for obvious space issues: in term of priority motivating a result comes before the discussion of its proof. However, we understand that our presentation regarding perturbation analysis may be improved and we thank R4 for pointing this out. We will also add some intuitions regarding this proof and its difficulties in the revision.

**Novelty of the discrete case. (R4)** R4 mentioned that "transition from continuous (gradient flow) to discrete (gradient descent) optimization – is also relatively simple". We are conscious that the transition from continuous to discrete may appear easy but we think that it is not an accident that the close related work only addressed the continuous dynamics: working on the discrete dynamics is more challenging. We are quite surprised that R4 do not mention at all the whole paragraph (L238-248) we wrote on "why the discrete analysis is challenging" where we developed some points to explain the new difficulties arising when working with the discrete dynamics. We explained in this paragraph why this transition is difficult. We encourage the reviewer to consider it carefully in their revision of the review.

**Implicit Regularization. (R4)** Regularization is a restriction within the search space of solution in order to improve generalization. A low rank constraint is an explicit regularization. Using a method that finds these low rank solutions without explicitly putting low rank constraint is a restriction in the search space of potential solutions with good generalization and thus is an implicit regularization. Early stopping is described as a regularization technique for deep learning in [Goodfellow, Bengio and Courville, 2016, §7.8] and is still relevant in practice, e.g. with corrupted labels [Li, Soltanolkotabi and Oymak, 2019].

We think that even though early stopping might not be necessary in some specific cases, the study of the optimization path is a conceptual advance in term of understanding of the inductive bias of gradient descent: it help to explain why the test 0-1 loss plateaus while the training optimization loss still decreases. For instance, in [Vaswani, Mishkin et al. 2019, Fig. 3] we can see that the 0-1 test accuracy plateaus while the training optimization loss is still decreasing showing that along the optimization path the solutions have at least as good generalization properties as the final solution.

**About initialization formula. (R1)** In Theorem 1, the matrix $Q$ can be chosen arbitrary. Thus, by density of the invertible matrices, for almost initialization $W_1$, one could find a matrix $Q$ to get the desired factorization. Thus, only $W_2$ requires to be specifically initialized. In practice, practitioners have the freedom to choose the initialization.

This initialization is necessary with the current proof technique (Lemma 5 in appendix is not true anymore if the $W_1$ and $W_2$ are not initialized with different $(\delta_i)_i$). We think that for almost all random initialization the phenomenon of sequential learning still occurs. It is confirmed by our experiments in §4.2 where $W_1$ and $W_2$ are initialized randomly.

Regarding the scaling, having different vanishing $\delta_i$ just rescales the times $T_i$ depending on the relative speed at which each component vanishes. For instance, if $\delta \to 0$, we would have $T_i = \delta_i/\delta\sigma_i$ and thus the phase transition time depends on the limit of the ratio $\delta/\delta_i$. We only presented the case $\delta_i = \delta$ for simplicity of the discussion.

[Meta-Review · NeurIPS 2019]

The paper studies the dynamics of discrete gradient descent for overparametrized two-layer neural networks and shows that under certain conditions on the input/output covariance matrices and the initialization the components of the input-output map are learned sequentially. The reviewers appreciated the contributions of the paper, both theory and experiments, and found the paper well written. At the same time, one reviewer feels the assumptions are too strong, and another one feels that some claims are misleading (e.g. having deep in the title) and that the contributions relative an un-cited paper by Lampinen and Ganguli are incremental. Post rebuttal, the reviewer concluded that the novelty of the paper is buried in the appendix, and that a re-write of the paper is needed to elucidate that novelty in the body of the paper. This AC agrees with R4 that the contributions relative to Lampinen and Ganguli need to be clearly established in the body of the paper and that a citation needs to be added. This AC also agrees that the title/abstract/body need to be changed to reflect that a shallow network with squared loss is being analyzed.